# Bi-Directional Communication-Efficient Stochastic FL via Remote Source Generation

**Maximilian Egger**\*, **Rawad Bitar, Antonia Wachter-Zeh**
Technical University of Munich
Munich, Germany
{maximilian.egger,rawad.bitar,antonia.wachter-zeh}@tum.de

**Nir Weinberger**
Israel Institute of Technology
Haifa, Israel
nirwein@technion.ac.il

**Deniz Gündüz**
Imperial College London
London, United Kingdom
d.gunduz@imperial.ac.uk

## Abstract

Federated Learning (FL) incurs high communication costs in both uplink and downlink. The literature largely focuses on lossy compression of model updates in deterministic FL. In contrast, stochastic (Bayesian) FL considers distributions over parameters, enabling uncertainty quantification, better generalization, and, crucially, inherent communication-regularized training through a mirror-descent structure. In this paper, we consider both uplink and downlink communication in stochastic FL, and propose a communication framework based on remote source generation. Employing Minimal Random Coding (MRC) for remote generation, we allow the server and the clients to sample from local and global posteriors (sources), respectively, rather than transmitting locally sampled updates. The framework encompasses communication-regularized local optimization and principled compression of model updates, leveraging gradually updated prior distributions as side information. Through extensive simulations, we show that our method achieves $5 - 32\times$ reduction in total communication cost while preserving accuracy. We further analyze the communication cost, refining existing MRC bounds and enabling precise quantification of uplink and downlink trade-offs. We also extend our method to conventional FL via stochastic quantization and prove a contraction property for the biased MRC compressor to facilitate convergence analysis.

## 1 Introduction

Federated learning (FL) enables collaborative machine learning (ML) across multiple clients orchestrated by a central federator [McMahan et al., 2017]. Communication efficiency, privacy, security, and data heterogeneity are well-established challenges in FL [Zhang et al., 2021, Wen et al., 2023]. As a bi-directional process, FL requires substantial *uplink* and *downlink* communication, posing increasing pressure on communication networks as ML models grow larger. To address this, lossy compression techniques have been widely adopted to reduce uplink gradient transmissions and downlink model broadcasts [Seide et al., 2014, Alistarh et al., 2017, Philippenko and Dieuleveut, 2020, Gruntkowska et al., 2023]. However, these methods almost exclusively focus on conventional (non-stochastic) FL, where clients train deterministic models and transmit fixed updates.

Alternatively, stochastic (Bayesian) FL offers improved generalization, robustness, and inherent uncertainty estimation [Zhang et al., 2022, Milasheuski et al., 2025]. Rather than training deterministic

---

\*Corresponding author

39th Conference on Neural Information Processing Systems (NeurIPS 2025).

models, clients learn local posterior distributions, aggregated by the federator to obtain a global posterior. Recently, [Isik et al., 2024] empirically demonstrated state-of-the-art performance under limited uplink bandwidth using stochastic compression methods, outperforming classical approaches. This framework can be applied to a variety of Bayesian FL solutions such as QSGD Alistarh et al. [2017], QLSD Vono et al. [2022], dithered quantization Abdi and Fekri [2019] and FedPM Isik et al. [2023], as well as to conventional FL settings augmented with stochastic compression.

A key technique enabling stochastic FL is *remote source generation*, which allows the federator to sample from the clients' local posterior, rather than obtaining samples locally generated by the clients. This avoids redundant transmission and enables tight, stochastic control over communication. Such remote generation requires common randomness shared between the transmitter and receiver in the form of a common prior Li [2024], which we also refer to as *side information*. If the downlink is unlimited, this allows the server to broadcast the global posterior to all the clients, and this posterior serves as a natural common prior. However, when both uplink and downlink are limited, the possibility for remote source generation is restricted, which challenges the application of efficient stochastic FL. Thus, in this paper, we explore and analyze communication-efficient stochastic FL. The rigorous treatment of stochastic FL in this scenario is further reinforced by its two fundamental advantages: (i) *Communication-regularized training*: we show that stochastic FL, as opposed to conventional solutions, inherently integrates communication constraints into the training process, effectively treating communication as an integral part of the optimization objective; (ii) *Priors as side information*: the probabilistic structure allows principled integration of common priors as side information, reducing communication costs to the update from prior to posterior.

Concretely, we address the following question: *Can joint uplink and downlink compression significantly reduce communication costs in stochastic FL without compromising accuracy?* We answer this question affirmatively, tightening the communication–accuracy trade-off in ways that deterministic methods cannot, and achieving **communication reductions of up to 32×** without performance loss. Below, we summarize our key contributions.

• We propose two novel bi-directional compression algorithms for stochastic FL with Minimal Random Coding (MRC): one leveraging globally shared randomness, and one requiring private shared randomness between each client and the federator. Both enable efficient sampling-based communication by exploiting carefully selected side information.

• We demonstrate substantial communication savings, reducing total cost by factors of $5 - 32$ while maintaining competitive accuracy across standard benchmarks. Our ablation studies analyze the effects of shared randomness and the choice of side information.

• We extend our approach to conventional FL with stochastic quantization, proving a contraction property of the resulting (biased) compression operator. This enables convergence guarantees in both uni- and bi-directional settings.

• We develop a theoretical framework for communication analysis in stochastic FL, quantifying uplink and downlink costs under MRC. Our results refine and generalize bounds from Chatterjee and Diaconis [2018], including tight analyses for Bernoulli distributions and tools applicable to broader distribution classes.

## 2 Preliminaries: Stochastic FL with Bi-Directional Compression

In this section, we shortly review the concepts of stochastic FL and compression based on MRC, which are employed in our proposed stochastic bi-directional algorithm.

**Stochastic FL.** A set of $n$ clients collaboratively and iteratively train a model, e.g., a neural network, under the orchestration of a federator. We primarily consider cross-silo federated learning with a moderate number of reliable clients [Kairouz and McMahan, 2021]. Client $i \in [n] := \{1, \ldots, n\}$ possesses a dataset $\mathcal{D}_i$. We differentiate between homogeneous data, where $\mathcal{D}_i$ is drawn independently from the same distribution for all clients (i.i.d.), and heterogeneous data, where each $\mathcal{D}_i$ may come from a different distribution (non i.i.d.). At each training iteration $t$, the federator holds a model $\theta_t$ described by a probability distribution. After downlink transmission, each client $i$ has an estimate $\hat{\theta}_{i,t}$ of $\theta_t$, and locally optimizes $\hat{\theta}_{i,t}$ to obtain a local probabilistic model, called *the posterior* $q_i^t$. Compressed clients' posteriors $q_i^t$ are transmitted back to the federator on the uplink to obtain an estimate $\hat{q}_i^t$. The federator aggregates the received posteriors using an aggregation rule $R(\cdot)$ to obtain

a refined global distribution $\theta_{t+1} = R\left(\{\hat{q}_i^t\}_{i \in [n]}\right)$. A simple aggregation rule $R(\cdot)$ is the average over all clients' posteriors. This process is repeated until a certain convergence criterion is met. In many stochastic FL settings, the sent client updates $\hat{q}_i^t$ are just samples from the posterior $q_i^t$.

In fact, conventional FL with stochastic quantization can also be described by the procedure above, though with the following differences: (i) the federator holds a model $\theta_t$ with deterministic parameters; (ii) each client $i$ locally optimizes $\hat{\theta}_{i,t}$ to obtain a local gradient $g_i^t$. A stochastic compression $Q_s(\cdot)$ is applied on the client's gradient to obtain a posterior distribution $q_i^t$ from $Q_s(g_i^t)$; (iii) samples of $q_i^t$ are transmitted to the federator on the uplink to obtain an estimate of the gradient, which we still denote by $\hat{q}_i^t$; and (iv) the federator updates the global model as $\theta_{t+1} = \theta_t - \eta R\left(\{\hat{q}_i^t\}_{i \in [n]}\right)$, with learning rate $\eta$. In this paper, we will investigate both settings, with a prominent focus on the former.

**Stochastic Compression by MRC.** To efficiently transmit samples from the posterior $q_i^t$, we employ MRC [Havasi et al., 2019], which allows to leverage shared randomness and side information common to the federator and the clients. MRC is a stochastic compressor $\mathcal{C}_{\mathrm{mrc}}(\cdot)$, whose input is a posterior distribution $Q$ and a prior distribution $P$, and its output is a sample from a distribution $\hat{Q}$ close to $Q$. It operates as follows: The encoder and decoder generate $n_{\mathrm{IS}}$ samples $\{X_i\}_{i \in [n_{\mathrm{IS}}]}$ from $P$. The encoder computes a categorical distribution $W$, with $W(i) \propto Q(X_i)/P(X_i)$, and transmits an index $i \sim W$ with $\log_2(n_{\mathrm{IS}})$ bits. To obtain high accuracy, it is required that $n_{\mathrm{IS}} = \Theta(\exp(\mathrm{D}_{\mathrm{KL}}(Q\|P)))$, where $\mathrm{D}_{\mathrm{KL}}(Q\|P)$ is the KL-divergence between $Q$ and $P$ [Chatterjee and Diaconis, 2018]. For brevity, in what follows, for two Bernoulli distributions with parameters $q$ and $p$, we will use the shorthands $\mathrm{d}_{\mathrm{KL}}(q\|p)$ and $\mathcal{C}_{\mathrm{mrc}}(q, p)$.

The choice of MRC for remote generation stems from a practicality aspect and ease of exposition. Ordered random coding (ORC) [Theis and Ahmed, 2022] is a natural advancement, reducing the entropy of the indices selected for transmission using the Gumbel-Max trick. The extension to ORC can be incorporated with our methods using a minor adaption that we omit here for clarity. Theis and Ahmed [2022] further expose an interesting connection to the Poisson functional representation, allowing exact remote generation. Such lossless sampling schemes, however, usually incur substantially larger communication costs and sampling complexities, often rendering those methods impractical. Thus, we herein focus on lossy remote generation schemes for efficiency.

# 3 BICOMPFL

We next introduce our method BICOMPFL, a bi-directional stochastic compression strategy, which uses MRC to reduce both uplink and downlink communication costs. The scheme assumes that shared randomness between each of the clients and the federator exists, which can be implemented using pseudo-random sequences generated from a common seed. We distinguish two types of shared randomness: private shared randomness (between individual clients and the federator) and global shared common randomness (among all parties), with the latter being more challenging to achieve in practice. We assume all clients and the federator share the same global model $\hat{\theta}_0$ at initialization. This does not incur any communication when global shared randomness is available, but necessitates an initial model transmission from the federator to clients when only private shared randomness exists.

**BICOMPFL: The General Algorithm.** Our method serves as a general framework for stochastic optimization procedures. We explain BICOMPFL for Bayesian FL and show in the sequel how it can be used for conventional FL with stochastic quantization. Consider probabilistic mask training (similar to FedPM, [Isik et al., 2023]) as an example of Bayesian FL. The models $\theta_t \in [0, 1]^d$ of dimension $d$ are parameters of Bernoulli distributions. Those parameters determine for each weight of a randomly initialized network with fixed weights $w$ whether it is activated or not. During inference, the weights $w$ are masked with samples $x^t \in \{0, 1\}^d \sim \theta_t$, i.e., the network weights are $w \odot x^t$. We start with a general description, which is valid for the cases of global and private shared randomness.

At iteration $t = 0$, each client $i \in [n]$ shares with the federator the same global model, i.e., $\hat{\theta}_{i,0} = \theta_0$, for all $i \in [n]$. At iteration $t$, each client $i$ locally trains model $\hat{\theta}_{i,t}$ in $L$ local iterations. In our previous example, when training Bernoulli distributions to mask a random network, the parameters are mapped to scores in a dual space, which are then trained for $L$ local iterations $m \in [L]$ using stochastic gradient descent. Mapping the trained scores back to the primal space, each client $i$ obtains a model update in terms of a posterior $q_i^t$. We refer to Appendix F for details. This optimization principle is a special instance of mirror descent, which, in the special case of optimizing over Bernoulli

---

**Algorithm 1** BICOMPFL-GR with Global Randomness

---

**Require:** Both clients and federator initialize the same global model $\theta_0$ using a shared seed
**Ensure:** Clients set prior $p^t = \hat{\theta}_{i,0} = \theta_0, \forall i \in [n]$

1: **repeat**
2:   **for** Client $i \in [n]$ **do**
3:     $q_i^t \leftarrow$ Local training of $\hat{\theta}_{i,t}$
4:     Employ $\mathcal{C}_{\mathrm{mrc}}(q_i^t, p^t)$ to sample indices $I_{i,\ell}^b, \ell \in [n_{\mathrm{UL}}], b \in [B]$ with prior $p^t$, transmitted to federator to reconstruct $\hat{q}_i^t$
5:   **end for**
6:   Federator updates global model $\theta_{t+1} = \frac{1}{n}\sum_{i=1}^n \hat{q}_i^t$
7:   Federator relays to client $j$ the other clients' indices $\{I_{i,\ell}^b\}_{\ell \in [n_{\mathrm{UL}}], b \in [B], i \in [n] \setminus \{j\}}$
8:   **for** Clients $i \in [n]$ **do**
9:     Reconstruct $\hat{\theta}_{i,t+1} = \frac{1}{n}\sum_{i=1}^n \hat{q}_i^t$ from $\{I_{i,\ell}^b\}$
10:   **end for**
11:   Clients and federator set prior $p^t = \hat{\theta}_{t+1}$
12:   $t \leftarrow t+1$
13: **until** Convergence

---

distributions, leads to a point-wise minimization with respect to a KL-proximity term (as opposed to the Euclidean distance in standard SGD, cf. Appendix C for details). The KL-divergence between the updated local model and the global model directly determines the communication cost. Hence, we *regularize* the minimization of the loss function by the communication cost, thereby treating communication as an inherent optimization objective.

To convey the model update $q_i^t$ to the federator, each client employs $\mathcal{C}_{\mathrm{mrc}}(\cdot)$ in $B$ blocks of size $d/B$ each (assuming for simplicity that $B$ divides $d$) with a prior distribution $p_{i,u}^t$, which is set to $p_{i,u}^0 = \hat{\theta}_{i,0}$ at iteration $t = 0$. The choice of $p_{i,u}^t$ for $t > 0$ will be clarified later. For each block $b \in [d/B]$, client $i$ conveys $n_{\mathrm{UL}}$ samples $\{y_{i,\ell}^t\}_{\ell \in [n_{\mathrm{UL}}]}$ of $q_i^t$ to the federator by transmitting for each block $b$ an index $I_{i,\ell}^b$ with $\log_2(n_{\mathrm{IS}})$ bits, where $n_{\mathrm{IS}}$ is the number of samples per block, generated from the prior distribution $p_{i,u}^t$ at both the client and the federator using the available shared randomness. The samples of all blocks are concatenated for each $\ell$. Hence, the federator obtains an estimate of client $i$'s posterior distribution using the empirical average $\hat{q}_i^t = \frac{1}{n_{\mathrm{UL}}}\sum_{\ell=1}^{n_{\mathrm{UL}}} y_{i,\ell}^t$.

By averaging the estimates $\hat{q}_i^t$ for all the clients' models, the federator updates the global model as $\theta_{t+1} = \frac{1}{n}\sum_{i=1}^n \hat{q}_i^t$. To transmit the new model to each client $i$, we assume the existence of a common prior $p_{i,d}^t$ shared by the federator and the clients. With $p_{i,d}^t$, the federator performs MRC in $B$ blocks of size $d/B$ to make client $i$ sample from, and thereby estimate, the latest global model $\theta_{t+1}$. The client samples $n_{\mathrm{DL}}$ masks $\{x_{i,\ell}^t\}_{\ell \in [n_{\mathrm{DL}}]}$, each incurring a communication cost of $B\log_2(n_{\mathrm{IS}})$ bits. An estimate of the updated global model is obtained by concatenating the reconstructed samples for all the blocks $b \in [B]$, and averaging over all masks $\hat{\theta}_{i,t+1} = \frac{1}{n_{\mathrm{DL}}}\sum_{\ell=1}^{n_{\mathrm{DL}}} x_{i,\ell}^t$.

Since the number of clients is typically large, $n_{\mathrm{UL}} = 1$ often suffices. The clients' contributions are averaged at the federator, effectively reducing the noise due to the MRC step. This allowed Isik et al. [2024] to theoretically analyze the uplink communication for importance sampling-based stochastic communication of model updates. We will follow a similar approach for downlink communication; however, since downlink communication cannot benefit from the averaging effect of multiple clients, we reduce the variance of the model estimate in the downlink by setting $n_{\mathrm{DL}} = n \cdot n_{\mathrm{UL}}$.

The choice of the priors $p_{i,u}^t$ and $p_{i,d}^t$ for MRC in the uplink and downlink channels, respectively, crucially affects the performance and the communication cost of the algorithm. As a first-order characterization, the communication cost of MRC is determined by $\mathrm{D_{KL}}(q_i^t \| p_{i,u}^t)$ in the uplink and by $\mathrm{D_{KL}}(\theta_{t+1} \| p_{i,d}^t)$ in the downlink. We continue the description with the easier setting in which global shared randomness is available, before turning to the more challenging setting of private randomness.

**Global Randomness.** When global shared randomness is available, all clients can maintain the same priors at each iteration $t$, and, thereby, obtain the same global model estimates $\hat{\theta}_{i,t}$. The global model is known to the clients and the federator from initialization, and synchronization among all clients

---

**Algorithm 2** BICOMPFL-PR with Private Randomness

---

**Require:** Both clients and federator initialize the same global model $\theta_0$ using a shared seed
**Ensure:** Clients set prior $p_{i,u}^t = p_{i,d}^t = \hat{\theta}_{i,0} = \theta_0, \forall i \in [n]$
 1: **repeat**
 2:   **for** Client $i \in [n]$ **do**
 3:     $q_i^t \leftarrow$ Local training of $\hat{\theta}_{i,t}$
 4:     Federator employs $\mathcal{C}_{\mathrm{mrc}}(q_i^t, p_{i,u}^t)$ to draw $n_{\mathrm{UL}}$ samples $y_{i,\ell}^t \sim q_i^t$ using prior $p_{i,u}^t$
 5:     Federator estimates client's posterior $\hat{q}_i^t = \frac{1}{n_{\mathrm{UL}}} \sum_{\ell=1}^{n_{\mathrm{UL}}} y_{i,\ell}^t$
 6:   **end for**
 7:   Federator updates global model $\theta_{t+1} = \frac{1}{n} \sum_{i=1}^n \hat{q}_i^t$
 8:   **for** Clients $i \in [n]$ **do**
 9:     Client employs $\mathcal{C}_{\mathrm{mrc}}(\theta_{t+1}, p_{i,d}^t)$ to draw $n_{\mathrm{DL}}$ samples $x_{i,\ell}^t \sim \theta_{t+1}$ using prior $p_{i,d}^t$
10:     Client est. global model: $\hat{\theta}_{i,t+1} = \frac{1}{n_{\mathrm{DL}}} \sum_{\ell=1}^{n_{\mathrm{DL}}} x_{i,\ell}^t$
11:     Clients set prior $p_{i,u}^t = p_{i,d}^t = \hat{\theta}_{i,t+1}$
12:   **end for**
13:   $t \leftarrow t + 1$
14: **until** Convergence

---

is ensured by choosing as prior $p_{i,u}^t = p_{i,d}^t$ the latest estimate of the global model $\hat{\theta}_{i,t}$. The clients utilize the globally shared randomness to sample the exact same samples from the same prior for uplink transmission at all iterations. Selected indices of such samples are transmitted to the federator to convey an estimate $\hat{q}_i^t$ of the posterior $q_i^t$, who reconstructs the global model $\theta_{t+1}$. Using the same prior in the downlink, i.e., the global model from the previous iteration, the updated model can be transmitted to the clients through MRC. Leveraging the shared randomness, all clients $i \in [n]$ sample from the same prior, and thus obtain the exact same estimate of the global model $\hat{\theta}_{i,t+1} = \hat{\theta}_{t+1}$, for all $i \in [n]$. Hence, we have that $p_{i,u}^t = p_{i,d}^t = \hat{\theta}_t$ for all $i \in [n]$.

In this version, the federator reconstructs the global model from estimates of the client posteriors $\hat{q}_i^t$. However, in the uplink, all clients sample from the same prior, which enables further improvements. Naively, the federator will reconstruct the global model using the indices $I_{i,\ell}^b$ for $b \in [B], \ell \in [n_{\mathrm{UL}}]$ received by the clients $i \in [n]$ through MRC, followed by an additional MRC round for downlink transmission. Instead, and more efficiently, the federator can simply relay the indices to the respective other clients (i.e., client $j$ receives $I_{i,\ell}^b$ for $b \in [B], i \in [n] \setminus \{j\}, \ell \in [n_{\mathrm{UL}}]$), which reconstruct the updated global model individually. This avoids additional noise by a second compression round and allows better convergence without additional communication facilitated by global randomness. We term this approach BICOMPFL-GR summarized in Algorithm 1.

**Private Randomness.** Without global randomness, maintaining the same prior among all clients is impossible without additional communication. Instead, an additional round of MRC is needed for the downlink transmission, and each client obtains a different estimate of the global model $\hat{\theta}_{i,t}$ at each iteration. Hence, the clients' local trainings start from different estimates of the global model. In a non-stochastic setting, this has only been considered by Philippenko and Dieuleveut [2021], Gruntkowska et al. [2024]. Understanding the additional cost incurred due to lack of shared randomness in terms of both the convergence speed, communication load, and the choice of the priors $p_{i,u}^t$ and $p_{i,d}^t$, is then of interest.

For the uplink transmission of client $i$, any convex combination of $\hat{\theta}_{i,t}$ and $\hat{q}_i^t$ can be used as prior, i.e., $p_{i,u}^t = \lambda \hat{\theta}_{i,t} + (1 - \lambda)\hat{q}_i^t$, for some $0 \leq \lambda \leq 1$ (cf. Appendix I.2 for details). This is due to the availability of both quantities at the federator and client $i$. However, small $\lambda$ values are not expected to reduce the cost of communication reflected by $d_{\mathrm{KL}}\left(q_i^t || p_{i,u}^t\right)$ since the previous global model estimate is likely to be similarly different from the posterior (in terms of the KL-divergence) than the previous posterior estimate of the federator. Indeed, our numerical experiments have shown that the savings from choosing $\lambda \neq 1$, i.e., priors other than $\hat{\theta}_{i,t}$, are not significant. For simplicity, we thus propose to use $p_{i,u}^t = p_{i,d}^t = \hat{\theta}_{i,t}$. We term this approach BICOMPFL-PR and summarize the procedure in Algorithm 2. Choosing different priors is possible and only affects line 11 in Algorithm 2. We mention in passing that BICOMPFL-PR allows partial client participation, which is incompatible

with shared randomness and the method BɪCᴏᴍᴘFL-GR. We further note that our methods are readily compatible with sparsification and pruning techniques (e.g., Wangni et al. [2018], Shi et al. [2019]) by excluding pruned parameters from the block selection.

**Block Allocation.** We consider three different block allocation strategies: 1) fixed block size (referred to as "Fixed" in the experiments), where each block $b \in [B]$ is of the same size and constant across all $t$; 2) adaptive block allocation (Adaptive) as proposed by Isik et al. [2024], where each block size is separately optimized each iteration $t$; and 3) adaptive average allocation (Adaptive-Avg), where the block sizes are equal but optimized at each iteration $t$ according to the average KL-divergence per block. We refer the reader to Appendix D for a detailed discussion on this.

**Partial Client Participation.** We note that BɪCᴏᴍᴘFL-PR supports partial client participation without additional communication overhead, as it does not rely on full prior synchronization among the clients. BɪCᴏᴍᴘFL-GR on the other hand, requires that all clients' priors are synchronized at each iteration. To allow partial client participation, the federator is required to transmit the previous global model to the clients that were absent in the previous iteration. This ensures full synchronization of the global model estimate used for efficient uplink communication. Hence, BɪCᴏᴍᴘFL-GR's compatibility with partial client participation comes with a potentially increased downlink communication cost for previously passive clients.

## 4 Experiments

We next evaluate the performance of our proposed BɪCᴏᴍᴘFL-GR and BɪCᴏᴍᴘFL-PR schemes in experiments, and compare against baseline FL strategies without compression (FedAvg or PSGD) [McMahan et al., 2017] and several non-stochastic bi-directional compression schemes that employ different combinations of compression, error-feedback, and momentum. In particular, we compare against DᴏᴜʙʟᴇSǫᴜᴇᴇᴢᴇ [Tang et al., 2019], Mᴇᴍ-SGD [Stich et al., 2018], Nᴇᴏʟɪᴛʜɪᴄ [Huang et al., 2022], CSᴇʀ [Xie et al., 2020], and the recently proposed Lɪᴇᴄ [Cheng et al., 2024]. SignSGD [Seide et al., 2014] serves to compress the transmitted gradients for all the schemes. We further compare with M3 [Gruntkowska et al., 2024], which partitions the model into disjoint parts for downlink transmission and transmits to each client a different part of the model. While M3 is focused on RandK compression for the uplink (i.e., transmitting random $K$ entries of the gradient), we use TopK [Wangni et al., 2018, Shi et al., 2019], which achieved much more stable results. As mentioned above, the mirror descent approach outlined in Section 3 inherently minimizes the communication cost as a by-product. Hence, it is a strong candidate for communication-efficient stochastic FL. Nonetheless, we show how our method can also be used to improve the communication efficiency in conventional FL, by using the uplink and downlink compression $\mathcal{C}_{\mathrm{mrc}}(\cdot)$ combined with stochastic quantizers, e.g., [Alistarh et al., 2017]. In Section 5, we pave the way to convergence guarantees by proving a contraction property of $\mathcal{C}_{\mathrm{mrc}}(\cdot)$ composed with a stochastic quantization $Q_s(\cdot)$ of gradients $g_i^t$. To compare our method to the baselines that use SignSGD as compressor, we evaluate BɪCᴏᴍᴘFL-GR in a conventional federated learning (CFL) task with a stochastic variant of SignSGD. We replace mirror descent over Bernoulli masks by a standard learning procedure over a deterministic model, which takes as input the global model estimate $\hat{\theta}_{i,t}$, computes a gradient $g_i^t$ (over $L$ local epochs, using SGD and cross-entropy losses), and outputs a distribution $Q_s(g_i^t)$.

In stochastic SignSGD, $Q_s(\cdot)$ transforms each gradient entry $g_{i,e}^t$ to a Bernoulli random variable with parameter $q_{i,e}^t = 1/(1 + \exp(-g_{i,e}^t/K))$ for some $K > 0$, where the random variable takes value $+1$ with probability $q_{i,e}^t$, and $-1$ otherwise. We then employ $\mathcal{C}_{\mathrm{mrc}}(q_i^t, p_{i,u}^t)$ to obtain samples $y_{i,\ell}^t$, where the compression is performed element-wise. We apply this method to BɪCᴏᴍᴘFL-GR where Step 6 is replaced by $\theta_{t+1} = \theta_t - \eta_s \frac{1}{n} \sum_{i=1}^n \hat{q}_i^t$, where $\hat{q}_i^t = \frac{1}{n_{\mathrm{UL}}} \sum_{\ell=1}^{n_{\mathrm{UL}}} y_{i,\ell}^t$ and $\eta_s$ is the federator's learning rate. Step 9 is modified accordingly. The priors $p^t$ are chosen as Bernoulli random variables with parameter $0.5$. We re-

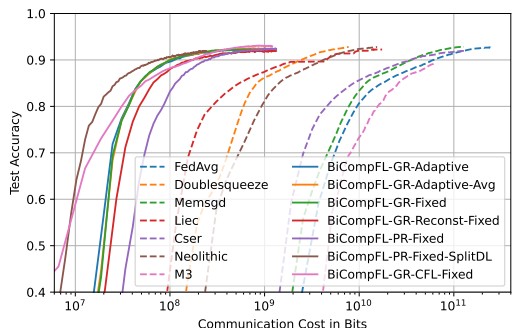

Figure 1: Test accuracy for BɪCᴏᴍᴘFL and baselines on Fashion MNIST 4CNN on i.i.d. data.

mark that while MRC samples are biased towards $p^t$ (as we discuss in Section 5), this particular prior choice avoids imbalance in stochastic SignSGD, and rather acts as a regularizer, pulling the clients' posteriors closer to maximum entropy distributions. Consequently, convergence is achieved under bi-directional compression even without error feedback. For general prior choices, error feedback may be needed, see Algorithm 3 and Appendix B. We will refer to this method as BICOMPFL-GR-CFL.

We study $n = 10$ clients (see Appendix H for additional experiments with more clients) collaboratively training a convolutional neural network (CNN)-based classifier for the datasets MNIST, Fashion-MNIST and CIFAR-10 under the orchestration of a federator. For MNIST, we use two different models, LeNet-5 [Lecun et al., 1998] and a 4-layer convolutional neural network (4CNN) proposed by Ramanujan et al. [2020]. The latter is also used to train on Fashion MNIST. For CIFAR-10, we use a larger neural network with 6 convolutional layers (6CNN). We train MNIST and Fashion-MNIST for 200 global iterations and CIFAR-10 for 400 global iterations. Through all experiments and datasets, we carry $L = 3$ local iterations per client. The learning rates are carefully selected to ensure convergence and comparability across all methods. Particularly, we tune the hyperparameters so that all algorithms achieve similar accuracies, allowing a fair comparison of their communication costs (see Appendix I.6 for details). Our main claims are the communication reduction of the bitrates per parameter per epoch, which are orthogonal to the choice of the learning rates of the algorithms. The code to reproduce our experiments is included in the supplementary material.

We evaluate the schemes in two settings: with a uniform data allocation (i.i.d.), to model homogeneous systems, and with a non-i.i.d. allocation, to model heterogeneous systems, where data allocation for each client is drawn from a Dirichlet distribution with parameter $\alpha = 0.1$. This regime is challenging due to extreme class imbalance. Each result shows the average across three simulation runs with different seeds. Further details on the simulation setup and the network architectures are deferred to Appendix E. Consistently throughout all experiments, our proposed methods provide **order-wise improvements** in the communication cost, while achieving state-of-the art accuracies.

We plot in Fig. 1 the test accuracies for all the schemes as a function of the total communication cost in bits per parameter and per global iteration. While all the schemes achieve approximately the same maximum test accuracy, BICOMPFL-GR and BICOMPFL-PR require substantially less communication. Hence, when the bandwidths of uplink and downlink transmissions are limited, both variations of the proposed method achieve better test accuracies. Turning our focus to the different variations of our scheme, it can be observed that, without partitioning the model for downlink compression, BICOMPFL-PR converges significantly slower than BICOMPFL-GR for any block allocation method. This highlights the intuition above that the additional MRC step in downlink incurs further noise, which reduces the convergence speed. However, when we partition the model in the downlink and only send disjoint parts to each client through MRC (BICOMPFL-PR-Fixed-SplitDL), the downlink communication cost reduces by a factor of $n$. In the regime of Fashion MNIST with a uniform data allocation, this comes without performance degradation, and is hence the method of choice in this regime. We additionally simulated BICOMPFL-GR with the suboptimal implementation (BICOMPFL-GR-Reconst-Fixed), in which the federator first reconstructs the global model, and then performs an additional MRC step for downlink transmission. This naturally reduces the convergence speed per iteration without gains in the communication cost. Hence, justifying the choice of BICOMPFL-GR. We show that, in conventional FL, BICOMPFL-GR-CFL substantially reduces the communication cost without loss in performance. In some cases, especially for non-i.i.d. data, we even observe improved performance, which we attribute to implicit regularization. Note that BICOMPFL-GR-CFL provides improvements even without error-feedback or momentum. However, our method is fully compatible with such techniques and can be used as a plug-in approach to further minimize the communication cost in many existing schemes. We study the convergence in Section 5.

We plot in Fig. 2(a) the schemes' average bitrates over the maximum test accuracy for MNIST and 4CNN. The average bitrate is reduced by more than a factor of 1000 compared to FedAvg, and more than a **factor of 32** compared to DOUBLESQUEEZE, NEOLITHIC and LIEC, which perform best among the conventional bi-directional compression methods. We repeat the study for non-i.i.d.data allocation according to a Dirichlet distribution with parameter $\alpha = 0.1$, and show maximum test accuracies over average bitrates in Fig. 2(b). Partitioning the model in BICOMPFL-PR worsens the final accuracy of the model. While the model converges faster, it does not achieve the same accuracies as BICOMPFL-GR and BICOMPFL-PR without partitioning. This hints towards hybrid schemes for BICOMPFL-PR, where the training begins with partitioning on the downlink, which is later switched

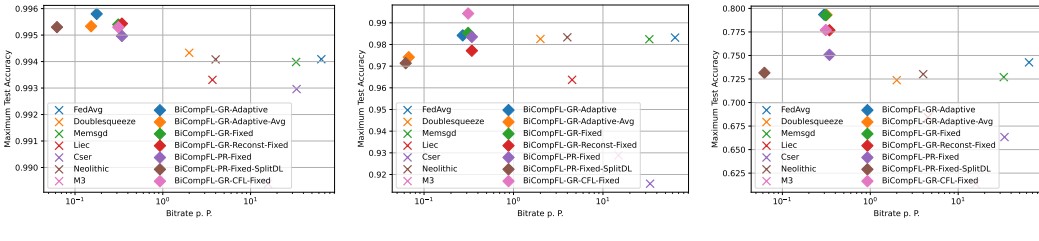

| | | | | |
|---|---|---|---|---|
| (a) MNIST 4CNN i.i.d. | | (b) MNIST 4CNN non-i.i.d. | | (c) CIFAR-10 6CNN i.i.d. |

Figure 2: Maximum test accuracy over total communication cost measured by bitrate per parameter.

to full transmission. In Fig. 2(c), we provide the results for CIFAR-10 and uniform data allocation. BICOMPFL-GR and BICOMPFL-PR both achieve better results with a bitrate **smaller by a factor of 5** than the best baselines. More detailed numerical results can be found in Appendices H and I.

The adaptive block allocation (Adaptive) of Isik et al. [2024] saves communication costs in many settings and provides better performance than the fixed block allocation (Fixed), due to more accurate MRC tailored to the exact divergences. The proposed low complexity adaptive strategy based on the average KL-divergence (Adaptive-Avg) per block can additionally save in communication (and computation) with no or little performance degradation. We refer the reader to Appendix H for further extensive experiments, graphs for accuracies over epochs, separate studies of uplink and downlink costs, and comparisons for the case of an available broadcast channel from federator to the clients. Finally, we refer to Appendix I for various ablation studies analyzing the sensitivity of BICOMPFL with respect to the choices of the priors, $n$, $n_{\text{DL}}$, $n_{\text{IS}}$, the block size $d/B$, and the learning rate $\eta$.

## 5 Theoretical Results

**Convergence.** In stochastic FL, the exact time dynamics of the system are challenging to analyze due to the round-dependent interplay of the learning procedure with the transmission noise. However, when using BICOMPFL for conventional FL with stochastic quantization (cf. BICOMPFL-GR-CFL), convergence guarantees can be given. We prove the convergence for a general and widely used class of stochastic quantizers $Q_s(\cdot)$, which are natively unbiased. $Q_s(\cdot)$ takes as input the entry $g_e$ of a gradient vector $\mathbf{g} \in \mathbb{R}^d$ and operates as follows. Let $s$ be the number of quantization intervals, and let $0 \leq \tau_e < s$ be an integer such that $\frac{\tau_e}{s} \leq \frac{|g_e|}{\|\mathbf{g}\|} \leq \frac{\tau_e+1}{s}$, then $Q_s(g_e)$ outputs $\|\mathbf{g}\| \cdot \text{sign}(g_e)(\tau_e + 1)/s$ with probability $s|g_e|/\|\mathbf{g}\| - \tau_e$, and $\|\mathbf{g}\| \cdot \text{sign}(g_e)\tau_e/s$ otherwise. $Q_s(\cdot)$ is unbiased, i.e., $\mathbb{E}[Q_s(\mathbf{x})] = \mathbf{x}$, and its variance satisfies $\mathbb{E}[\|Q_s(\mathbf{x}) - \mathbf{x}\|^2] \leq \min\{d/s^2, \sqrt{d}/s\}\|\mathbf{x}\|_2^2$ [Alistarh et al., 2017]. See Remark 2 in Appendix B for a discussion on the choice of $s$.

Replacing stochastic SignSGD by $Q_s(\cdot)$ in BICOMPFL-GR-CFL, the posterior is given by a Bernoulli distribution with parameter $q_{i,e}^t = s|g_{i,e}^t|/\|g_i^t\| - \tau_e$. The values $\|\mathbf{g}\|$, $\text{sign}(\mathbf{g})$, and $\tau_e$ can be encoded independently, e.g., using Elias coding. With a slight abuse of notation, let $\mathcal{C}_{\text{mrc}}(Q_s(\cdot), \cdot)$ denote the composition of $Q_s(\cdot)$ and MRC with $n_{\text{IS}}$ samples per entry. The compression $\mathcal{C}_{\text{mrc}}(Q_s(g_i^t), \cdot)$ takes a gradient $g_i^t$ and outputs samples from a distribution close to $Q_s(g_i^t)$, and falls in the class of biased compressors. We can prove the following contraction property for $\mathcal{C}_{\text{mrc}}(Q_s(\cdot), \cdot)$, which will facilitate convergence analysis for uni- and bi-directional compression. This constitutes a substantial improvement over [Isik et al., 2024], where such guarantees were missing, and hence no convergence guarantees were given. A prominent biased contractive compressor is TopK.

**Lemma 1.** *For any $\mathbf{x} \in \mathbb{R}^d$ and corresponding posterior $q$ following $Q_s(\mathbf{x})$, and a prior $p \in [0, 1]^d$, let $\bar{\Delta} := \max_{e \in [d]} \frac{q_e}{p_e} - \frac{1-q_e}{1-p_e}$, $\bar{\Delta}' := \max_{e \in [d]} q_e \left(\frac{p_e}{q_e} + \frac{1-p_e}{1-q_e}\right)$, and $\bar{p} := \max_{e \in [d]} p_e$. The compressor $\mathcal{C}_{\text{mrc}}(Q_s(\cdot))$ satisfies the following contraction property for $n_{\text{IS}} = \mathcal{O}(\max\{\sqrt{2\bar{\Delta}'}, \log(6\bar{p}(\bar{\Delta} + \bar{\Delta}^2))\sqrt{6\bar{p}(\bar{\Delta} + \bar{\Delta}^2)}\})$ and $s \geq \sqrt{2d}$:*

$$\mathbb{E}[\|\mathcal{C}_{\text{mrc}}(Q_s(\mathbf{x})) - \mathbf{x}\|^2] \leq (1 - \delta)\|\mathbf{x}\|^2,$$

*for $\delta = 1 - \frac{d}{s^2}\left(1 + \frac{\bar{\Delta}'}{n_{\text{IS}}^2} + \mathcal{O}\left((\bar{\Delta} + \bar{\Delta}^2)\sqrt{\frac{6\bar{p}\log(2n_{\text{IS}})}{n_{\text{IS}}}}\right)\right).$*

The underlying core result is a refinement of the MRC analysis, cf. Lemma 2 (Appendix A). Hence, for sufficiently large $n_{\text{IS}}$, the compressor $\mathcal{C}_{\text{mrc}}(Q_s(\cdot), \cdot)$ can be used as an alternative to common compressors such as $Q_s(\cdot)$. The use of MRC introduces a bias into the otherwise unbiased stochastic quantization. Based on the contraction property in Lemma 1, standard convergence results (cf. Theorem 2) follow easily by a straightforward extension of our conventional FL algorithm BICOMPFL-GR-CFL to error feedback (cf. Algorithm 3) as detailed in Appendix B.

**Communication Cost.** We analyze the communication cost in a specific iteration $t$ and comment on the inter-round dependency later. When the latest global model estimate $\hat{\theta}_{i,t}$ is chosen as a prior in MRC, the uplink cost is determined by how far the model evolves during the client's training, i.e., $\text{d}_{\text{KL}}(q_i^t || p_{i,u}^t) = \text{d}_{\text{KL}}(q_i^t || \hat{\theta}_{i,t})$. After communicating samples of the posteriors, the federator obtains an estimate $\hat{q}_i^t$ for all $i \in [n]$. The cost of communication on the downlink to client $i$ is then determined by $\text{d}_{\text{KL}}(\frac{1}{n}\sum_{i=1}^{n}\hat{q}_i^t || \hat{\theta}_{i,t})$. While $\text{d}_{\text{KL}}(q_i^t || \hat{\theta}_{i,t})$ depends on the progress during client training, the core challenge is to bound the expected KL-divergence of each model estimate $\text{d}_{\text{KL}}(\hat{q}_i^t || \hat{\theta}_{i,t})$ in the presence of potentially different priors, i.e., $\hat{\theta}_{i,t} \neq \hat{\theta}_{j,t}, i \neq j$. For each client $i$, the overall communication cost is in the order of $n_{\text{DL}}\exp\left(\text{d}_{\text{KL}}(\frac{1}{n}\sum_{i=1}^{n}\hat{q}_i^t || p_{i,d}^t)\right) + n_{\text{UL}}\exp\left(\text{d}_{\text{KL}}(q_i^t || p_{i,u}^t)\right)$. We will next quantify $\text{d}_{\text{KL}}(\frac{1}{n}\sum_{i=1}^{n}\hat{q}_i^t || \hat{\theta}_{i,t})$ for the case $p_{i,u}^t = p_{i,d}^t$, however, the analysis can be extended to $p_{i,u}^t \neq p_{i,d}^t$ by an additional assumption on the divergence between the two priors.

For the theoretical analysis, we focus on the scalar case for a single iteration $t$, where client $i \in [n]$ has a posterior $Q_i$, and the federator and client $i$ share a common prior $P_i$, both are Bernoulli distributions with parameters $q_i$ and $p_i$, respectively. In the context of FL, the client locally trains $P_i$ and results with $Q_i$. According to Chatterjee and Diaconis [2018] and the multi-client extension of Isik et al. [2024], the communication cost in the uplink is determined by $\exp(\text{d}_{\text{KL}}(Q_i || P_i))$. After uplink transmission, the federator obtains an estimate $\hat{q}_i$ of $q_i$; and hence, the updated global model is given by $\frac{1}{n}\sum_{i=1}^{n}\hat{q}_i$. The downlink cost for client $i$ is determined by $\text{d}_{\text{KL}}\left(\frac{1}{n}\sum_{i=1}^{n}\hat{q}_i || p_i\right)$.

We derive a new high probability upper bound on this quantity, refining previous MRC analysis for the special case of Bernoulli distributions. While a more general analysis can be conducted for other classes of distributions (cf. Remark 1 in Appendix A), the Bernoulli-based optimization method described earlier proves particularly efficient by enabling communication-regularized training, a unique property that renders our methods substantially more communication-efficient while preserving state-of-the-art performance.

Let $X$ be a Bernoulli sample obtained through MRC. As an initial step, we bound the difference between $q_i$ and the probability $\Pr(X = 1)$ that the samples are drawn from, which vanishes when $p_i = q_i$ (and hence $\text{d}_{\text{KL}}(q_i || p_i) = 0$). We note that the bound of Chatterjee and Diaconis [2018, Theorem 1.1] does not satisfy this natural property. We formally state the result in Proposition 1 in Appendix A, which, however, does not yet capture the dependency on the number of samples $n_{\text{IS}}$ used in MRC to sample an index. We refine Proposition 1 with Lemma 2 (cf. Appendix A), which additionally captures this dependency, and will allow us to derive an upper bound on $\text{d}_{\text{KL}}\left(\frac{1}{n}\sum_{i=1}^{n}\hat{q}_i || p_i\right)$. Lemma 2 is of independent interest and can be seen as a refinement of [Chatterjee and Diaconis, 2018] for Bernoulli distributions. It is required to prove Theorem 1.

For the statement of the following theorem, we assume that the progress by one local client training is bounded by $|q_j - p_j| \leq \rho$ for all $j \in [n]$. Using Pinsker's inequality to bound $|q_j - p_j| \leq \frac{1}{2}\sqrt{\text{d}_{\text{KL}}(q_j || p_j)/2}$, this is a natural assumption given from the KL-proximity term of mirror descent (for one local iteration), and can be strictly enforced through the projection of $q_j$ onto a KL ball around $p_j$ of fixed divergence. We assume that the difference between the clients' priors, i.e., their global model estimates in our algorithms, are bounded as $|p_i - p_j| \leq \zeta$ for all $i, j \in [n]$.

**Theorem 1.** *Assume $p_j > \zeta$ for all $j \in [n]$, for $\Delta_j := \frac{q_j}{p_j - \zeta} - \frac{1-q_j}{1-p_j+\zeta}$ and $\Delta_j' := q_j\left(\frac{p_j+\zeta}{q_j} + \frac{1-p_j+\zeta}{1-q_j}\right)$, with probability $1 - \delta'$, the global model divergence $d_{KL}(\frac{1}{n}\sum_{j=1}^{n}\hat{q}_j || p_i)$ is upper bounded by*

$$\sum_{j=1}^{n}\frac{2}{n\min\{p_i, 1-p_i\}}\left(\frac{\Delta_j'}{n_{IS}^2} + \sqrt{\frac{\ln(2/\delta')}{2n_{UL}}} + \rho + \zeta^2 + \mathcal{O}\left((\Delta_j + \Delta_j^2)\sqrt{\frac{6(p_i + \zeta)\log(2n_{IS})}{n_{IS}}}\right)\right).$$

By Chatterjee and Diaconis [2018], this provides an immediate bound on the cost of downlink transmission. The bound applies to both algorithms BICOMPFL-PR and BICOMPFL-GR. However, when all priors $p_j$ are the same (such as in BICOMPFL-GR-Reconst), i.e., $\zeta = 0$, the bound simplifies

accordingly. The explicit dependency on the factor $1/\sqrt{n_{\mathrm{UL}}}$ reflects the interplay between uplink and downlink cost. The parameter $\zeta$ gives rise to an inter-round dependency of the communication cost. The more accurate the estimation of the global model in the previous iteration (given the priors are chosen as $\hat{\theta}_{i,t}$), the smaller $\zeta$, and hence the lower the transmission cost in the subsequent iteration. The proofs of Proposition 1, Lemma 2, and Theorem 1 can be found in Appendix A.

## 6  Related Work

Following the introduction of FL [McMahan et al., 2017], lossy compression of gradients or model updates has been a long-studied narrative in FL, with prominent representatives such as SignSGD, also known as 1-bit Stochastic Gradient Descent (SGD) [Seide et al., 2014], QSGD [Alistarh et al., 2017], TernGrad [Wen et al., 2017], SignSGD with error feedback [Karimireddy et al., 2019], vector-quantized SGD [Gandikota et al., 2021] and natural compression [Horvóth et al., 2022]. Such methods retain satisfactory final model accuracy even with aggressive quantization. Sparsification-based methods have also been considered as alternatives, e.g., TopK [Wangni et al., 2018, Shi et al., 2019]. The importance of bi-directional gradient compression in many settings was outlined by Philippenko and Dieuleveut [2020]. Many schemes were proposed that leverage combinations of gradient compression in the uplink and downlink, error-feedback, and momentum, e.g., Mem-SGD [Stich et al., 2018], DoubleSqueeze [Tang et al., 2019], block-wise SignSGD with momentum [Zheng et al., 2019], communication-efficient SGD with error reset (Cser) [Xie et al., 2020], Artemis [Philippenko and Dieuleveut, 2020], Neolithic [Huang et al., 2022], DoCoFL [Dorfman et al., 2023], EF21-P and friends [Gruntkowska et al., 2023], 2Direction [Tyurin and Richtárik, 2023], M3 [Gruntkowska et al., 2024], and LIEC [Cheng et al., 2024]. With the exception of the methods MCM [Philippenko and Dieuleveut, 2021] and M3 [Gruntkowska et al., 2024], each client receives the same broadcast, potentially compressed, global gradient or model update. Isik et al. [2024] studied uplink compression for stochastic FL and showed significant communication reduction with competitive performance. Their framework, termed KLMS, applies to a variety of stochastic compressors and to Bayesian FL settings, e.g., QLSD Vono et al. [2022]. The compression is based on importance sampling and MRC, thoroughly studied by Chatterjee and Diaconis [2018] and Havasi et al. [2019]. Such methods, known as relative entropy coding, have been used in FL in conjunction with differential privacy, cf. DP-REC [Triastcyn et al., 2022].

Since the lottery ticket hypothesis [Frankle and Carbin, 2019], finding sparse subnetworks of neural networks that achieve satisfactory accuracy was investigated. Ramanujan et al. [2020] showed that randomly weighted networks contain suitable subnetworks of large neural networks capable of achieving competitive performance. Isik et al. [2023] formulated a probabilistic method of training neural network masks collaboratively in an FL context.

## 7  Conclusion

We illuminated bi-directional compression in stochastic FL via federated probabilistic mask training, which we showed to inherently optimize both the learning objective and the communication costs. By leveraging side information through carefully chosen prior distributions, the total communication costs are reduced by factors between $5-32$ compared to non-stochastic FL baselines, while achieving state-of-the-art accuracies on classification tasks, for both homogeneous and heterogeneous data. We thus close the gap of downlink compression for stochastic FL and complement the existing literature on bi-directional compression for standard FL. Applying our methods to stochastic quantization in conventional FL, we paved the way to convergence analysis for MRC-based compression. Allowing different priors among all clients, this work opens the door to studying compression under side-information in *decentralized stochastic FL*, where a central coordinator is missing. Our theoretical results are of independent interest and may be applied in various scenarios where MRC is used.

Client privacy and fairness are important directions beyond the primary focus of this work. Compression methods commonly strengthen the clients' privacy in FL. Particularly, the noise introduced through our sampling methods dilute the individual contributions of the clients observed by the federator, naturally enhancing the clients' privacy and reducing the risk of privacy breaches through, e.g., membership inference attacks. Quantifying the privacy-utility trade-off that arises in the presence of bi-directional compression is left as an interesting direction for future work.

## Acknowledgments

The research of R. Bitar is supported by the German Research Foundation (DFG) under grant agreement number BI 2492/1-1. The research of N. Weinberger was partially supported by the Israel Science Foundation (ISF), grant no. 1782/22. D. Gündüz acknowledges funding from UKRI under the ERC-Consolidator project AI-R (EP/X030806/1) and INFORMED-AI Hub (EP/Y028732/1).

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

## A  Proofs and Intermediate Results

In the following, we provide the formal statements of Proposition 1 and Lemma 2 including their proofs. Parts of the proof of Proposition 1 will be used to prove Lemma 2. We prove Theorem 1 afterward.

**Proposition 1.** *For a sample $X_\ell$ transmitted by MRC with posterior and prior Bernoulli distributions with parameters $q$ and $p$, we have*

$$|\Pr(X_\ell = 1) - q| \le q \left( \max\left\{ \frac{p}{q}, \frac{1-p}{1-q}, \frac{q}{p}, \frac{1-q}{1-p} \right\} - 1 \right).$$

*Proof of Proposition 1.* Assume a party wants to sample from a Bernoulli distribution $Q$ with parameter $q$, which is held by another party. Both parties share a common prior $P$ in the form of a Bernoulli distribution with parameter $p$ and have access to shared randomness. Fix any sample index $\ell$ for the moment (this index will be needed for the proof of Theorem 1). Both parties sample $Kn_{\mathrm{IS}}$ i.i.d. samples $X_{\ell,i} \sim P$ for $i \in [n_{\mathrm{IS}}]$ independently and identically from $P$. The party holding $Q$ constructs an auxiliary distribution

$$W_\ell(i) = \frac{Q(X_{\ell,i})/P(X_{\ell,i})}{\sum_{i=1}^{n_{\mathrm{IS}}} Q(X_{\ell,i})/P(X_{\ell,i})},$$

from which it samples to obtain an index $I_\ell$. The index is transmitted to the other party, which reconstructs the corresponding sample $X_{\ell,I_\ell}$.

To bound the difference $|\Pr(X_\ell = 1) - q|$, i.e., the target Bernoulli parameter compared to the parameter which the sample is drawn from, by the independence of the samples $X_{\ell,I_\ell}$ for different $\ell$,

we focus on a single sample $\ell \in [K]$, for which it holds that

$$\Pr(X_{\ell,I_\ell} = 1)$$

$$= \sum_{i=1}^{n_{\mathrm{IS}}} \sum_{\{x_1,\dots,x_{n_{\mathrm{IS}}}:x_i=i\}} \Pr(X_{\ell,1} = x_1,\dots,X_{\ell,n_{\mathrm{IS}}} = x_{n_{\mathrm{IS}}}) \Pr(I_\ell = i \mid X_{\ell,1} = x_1,\dots,X_{\ell,n_{\mathrm{IS}}} = x_{n_{\mathrm{IS}}})$$

$$\overset{(a)}{=} n_{\mathrm{IS}} \sum_{\{x_2,\dots,x_{n_{\mathrm{IS}}}\}} \Pr(X_{\ell,1} = 1, X_{\ell,2} = x_2,\dots,X_{\ell,n_{\mathrm{IS}}} = x_{n_{\mathrm{IS}}})$$

$$\cdot \Pr(I_\ell = 1 \mid X_{\ell,1} = 1, X_{\ell,2} = x_2,\dots,X_{\ell,n_{\mathrm{IS}}} = x_{n_{\mathrm{IS}}})$$

$$\overset{(b)}{=} n_{\mathrm{IS}} \sum_{\mathrm{L}=0}^{n_{\mathrm{IS}}-1} \sum_{\{x_2,\dots,x_{n_{\mathrm{IS}}}:\sum_{i=2}^{n_{\mathrm{IS}}}=\mathrm{L}\}} \Pr(X_{\ell,1} = 1, X_{\ell,2} = x_2,\dots,X_{\ell,n_{\mathrm{IS}}} = x_{n_{\mathrm{IS}}})$$

$$\cdot \Pr(I_\ell = 1 \mid X_{\ell,1} = 1, X_{\ell,2} = x_2,\dots,X_{\ell,n_{\mathrm{IS}}} = x_{n_{\mathrm{IS}}}),$$

where $(a)$ follows from symmetry, $(b)$ follows since by permutation invariance, the inner probability only depends on the number of ones in $\{x_2,\dots,x_{n_{\mathrm{IS}}}\}$.

The inner probability is given by the distribution $W_\ell(i)$. Given that $X_{\ell,1} = 1$ and that $\sum_{i=2}^{n_{\mathrm{IS}}} X_{\ell,\ell} = \mathrm{L}$, it holds that

$$\sum_{i=1}^{n_{\mathrm{IS}}} Q(X_{\ell,i})/P(X_{\ell,i}) = (\mathrm{L}+1) \cdot \frac{q}{p} + (n_{\mathrm{IS}} - \mathrm{L} - 1) \cdot \frac{1-q}{1-p}.$$

Hence,

$$\Pr(I_\ell = 1 \mid X_{\ell,1} = 1, X_{\ell,2} = x_2,\dots,X_{\ell,n_{\mathrm{IS}}} = x_{n_{\mathrm{IS}}}) = \frac{\frac{q}{p}}{(\mathrm{L}+1) \cdot \frac{q}{p} + (n_{\mathrm{IS}} - \mathrm{L} - 1) \cdot \frac{1-q}{1-p}},$$

which is independent of the exact choice of $\{x_2,\dots,x_{n_{\mathrm{IS}}}\}$ given their sum $\sum_{i=2}^{n_{\mathrm{IS}}} X_{\ell,i} = \mathrm{L}$. Since $\Pr(X_{\ell,1} = 1, X_{\ell,2} = x_2,\dots,X_{\ell,n_{\mathrm{IS}}} = x_{n_{\mathrm{IS}}}) = p^{\mathrm{L}+1}(1-p)^{n_{\mathrm{IS}}-\mathrm{L}-1}$ by the Bernoulli distribution assumption, we have

$$\Pr(X_{\ell,I_\ell} = 1) = n_{\mathrm{IS}} \sum_{\mathrm{L}=0}^{n_{\mathrm{IS}}-1} \binom{n_{\mathrm{IS}}-1}{\mathrm{L}} p^{\mathrm{L}+1}(1-p)^{n_{\mathrm{IS}}-\mathrm{L}-1} \frac{\frac{q}{p}}{(\mathrm{L}+1) \cdot \frac{q}{p} + (n_{\mathrm{IS}} - \mathrm{L} - 1) \cdot \frac{1-q}{1-p}},$$

Defining a binary random variable M with sample space $\left\{\frac{q}{p}, \frac{1-q}{1-p}\right\}$, for a Bernoulli distribution $\mathrm{Ber}\left(\frac{\mathrm{L}+1}{n_{\mathrm{IS}}}\right)$ with success probability parameter $\frac{\mathrm{L}+1}{n_{\mathrm{IS}}}$, where a success refers to the outcome $\mathrm{M} = \frac{q}{p}$, we can write that

$$\Pr(X_{\ell,I_\ell} = 1) = q \cdot \sum_{\mathrm{L}=0}^{n_{\mathrm{IS}}-1} \binom{n-1}{\mathrm{L}} p^{\mathrm{L}}(1-p)^{n_{\mathrm{IS}}-\mathrm{L}-1} \frac{1}{\frac{\mathrm{L}+1}{n_{\mathrm{IS}}}\frac{q}{p} + \frac{n_{\mathrm{IS}}-\mathrm{L}-1}{n_{\mathrm{IS}}}\frac{1-q}{1-p}}$$

$$= q \cdot \mathbb{E}\left[\frac{1}{\frac{\mathrm{L}+1}{n_{\mathrm{IS}}}\frac{q}{p} + \frac{n_{\mathrm{IS}}-\mathrm{L}-1}{n_{\mathrm{IS}}}\frac{1-q}{1-p}}\right] = q\mathbb{E}\left[\frac{1}{\mathbb{E}_{\mathrm{Ber}\left(\frac{\mathrm{L}+1}{n_{\mathrm{IS}}}\right)}[\mathrm{M}]}\right] \qquad (1)$$

$$\overset{(a)}{\leq} q\mathbb{E}\left[\mathbb{E}_{\mathrm{Ber}\left(\frac{\mathrm{L}+1}{n_{\mathrm{IS}}}\right)}\left[\frac{1}{\mathrm{M}}\right]\right],$$

where the outer expectation is over the binomial distribution with $n_{\mathrm{IS}} - 1$ trials and success probability $p$, i.e., $\mathrm{L} \sim \mathrm{Binomial}(n_{\mathrm{IS}} - 1, p)$, and where $(a)$ follows from Jensen's inequality over the inner expectation. Hence,

$$\Pr(X_{\ell,I_\ell} = 1) - q = q\left(\frac{\Pr(X_{\ell,I_\ell} = 1)}{q} - 1\right)$$

$$\leq q\left(\mathbb{E}\left[\mathbb{E}_{\mathrm{Ber}\left(\frac{\mathrm{L}+1}{n_{\mathrm{IS}}}\right)}\left[\frac{1}{\mathrm{M}}\right]\right] - 1\right) \qquad (2)$$

Since $\frac{1}{\mathbb{E}_{\mathrm{Ber}\left(\frac{L+1}{n_{\mathrm{IS}}}\right)}[\mathrm{M}]} \geq 2 - \mathbb{E}_{\mathrm{Ber}\left(\frac{L+1}{n_{\mathrm{IS}}}\right)}[\mathrm{M}]$, it also follows from (1) that

$$\Pr(X_{\ell,I_\ell} = 1) = q \cdot \mathbb{E}\left[\frac{1}{\frac{L+1}{n_{\mathrm{IS}}}\frac{q}{p} + \frac{n_{\mathrm{IS}}-L-1}{n_{\mathrm{IS}}}\frac{1-q}{1-p}}\right] = q\mathbb{E}\left[\frac{1}{\mathbb{E}_{\mathrm{Ber}\left(\frac{L+1}{n_{\mathrm{IS}}}\right)}[\mathrm{M}]}\right]$$

$$\geq q\mathbb{E}\left[2 - \mathbb{E}_{\mathrm{Ber}\left(\frac{L+1}{n_{\mathrm{IS}}}\right)}[\mathrm{M}]\right],$$

from which we have

$$\Pr(X_{\ell,I_\ell} = 1) - q \geq q\left(1 - \mathbb{E}\left[\mathbb{E}_{\mathrm{Ber}\left(\frac{L+1}{n_{\mathrm{IS}}}\right)}[\mathrm{M}]\right]\right). \tag{3}$$

Combining the upper and lower bound in (2) and (3), respectively, we derive

$$|\Pr(X_{\ell,I_\ell} = 1) - q| \leq q\left(\max\left\{\mathbb{E}\left[1 - \mathbb{E}_{\mathrm{Ber}\left(\frac{L+1}{n_{\mathrm{IS}}}\right)}[\mathrm{M}]\right], \mathbb{E}\left[\mathbb{E}_{\mathrm{Ber}\left(\frac{L+1}{n_{\mathrm{IS}}}\right)}\left[\frac{1}{\mathrm{M}}\right]\right]\right\} - 1\right)$$

$$\leq q\left(\mathbb{E}\left[\max\left\{\mathbb{E}_{\mathrm{Ber}\left(\frac{L+1}{n_{\mathrm{IS}}}\right)}[\mathrm{M}], \mathbb{E}_{\mathrm{Ber}\left(\frac{L+1}{n_{\mathrm{IS}}}\right)}\left[\frac{1}{\mathrm{M}}\right]\right\}\right] - 1\right)$$

$$\leq q\left(\mathbb{E}\left[\mathbb{E}_{\mathrm{Ber}\left(\frac{L+1}{n_{\mathrm{IS}}}\right)}\left[\max\left\{\mathrm{M}, \frac{1}{\mathrm{M}}\right\}\right]\right] - 1\right)$$

$$\leq q\left(\mathbb{E}\left[\max\left\{\frac{p}{q}, \frac{1-p}{1-q}, \frac{q}{p}, \frac{1-q}{1-p}\right\}\right] - 1\right)$$

$$= q\left(\max\left\{\frac{p}{q}, \frac{1-p}{1-q}, \frac{q}{p}, \frac{1-q}{1-p}\right\} - 1\right).$$

This concludes the proof. $\qquad\square$

**Lemma 2.** *For a sample $X_\ell$ transmitted via MRC with posterior and prior being Bernoulli distributions with parameters $q$ and $p$, $\Delta := \frac{q}{p} - \frac{1-q}{1-p}$ and $\Delta' := q\left(\frac{p}{q} + \frac{1-p}{1-q}\right)$, we have*

$$|\Pr(X_\ell = 1) - q| \leq \frac{\Delta'}{n_{IS}^2} + \mathcal{O}\left((\Delta + \Delta^2)\sqrt{\frac{6p\log(2n_{IS})}{n_{IS}}}\right).$$

*Proof of Lemma 2.* The proof starts with the same derivations as for the proof of Proposition 1, which we follow until (1) to get

$$\Pr(X_{\ell,I_\ell} = 1) = q\mathbb{E}\left[\frac{1}{\mathbb{E}_{\mathrm{Ber}\left(\frac{L+1}{n_{\mathrm{IS}}}\right)}[\mathrm{M}]}\right]$$

Since L is a random quantity that follows a Binomial distribution, we bound $|\Pr(X_{\ell,I_\ell} = 1) - q|$ using a concentration bound on L. The relative (multiplicative) Chernoff bound states that

$$\Pr(|L - \varepsilon(n_{\mathrm{IS}}p)| \geq \varepsilon n_{\mathrm{IS}}p) = \Pr(L - \varepsilon(n_{\mathrm{IS}}p) \geq \varepsilon n_{\mathrm{IS}}p) + \Pr(L - \varepsilon(n_{\mathrm{IS}}p) \leq -\varepsilon n_{\mathrm{IS}}p)$$

$$\leq 2\exp\left(-\frac{\varepsilon^2 n_{\mathrm{IS}}p}{3}\right)$$

for any $\varepsilon \in [0,1]$. Setting $\varepsilon = \sqrt{\frac{3\log(2/\delta)}{n_{\mathrm{IS}}p}}$ implies that

$$|L - n_{\mathrm{IS}}p| \geq \sqrt{3n_{\mathrm{IS}}p\log(2/\delta)}$$

with probability at most $\delta$. Setting $\delta = \frac{1}{n_{\mathrm{IS}}^2}$, we obtain for a concentration parameter[2] $\eta_\delta := \sqrt{\frac{6p\log(2n_{\mathrm{IS}})}{n_{\mathrm{IS}}}}$ that

$$\mathcal{E} := \{|L - n_{\mathrm{IS}}p| \geq n_{\mathrm{IS}}\eta_\delta\}$$

---

[2]Note that we can assume $p + \eta_\delta \leq 1$ and $p - \eta_\delta \geq 0$, otherwise the concentration can be trivially bounded.

with probability $\Pr(\mathcal{E}) \leq \frac{1}{n_{\mathrm{IS}}^2}$.

Then, we can write

$$\Pr(X_{\ell,I_\ell} = 1) = q\mathbb{E}\left[\frac{1}{\mathbb{E}_{\mathrm{Ber}\left(\frac{L+1}{n_{\mathrm{IS}}}\right)}[\mathrm{M}]}\right]$$

$$= q\mathbb{E}\left[\frac{1}{\mathbb{E}_{\mathrm{Ber}\left(\frac{L+1}{n_{\mathrm{IS}}}\right)}[\mathrm{M}]} \cdot \mathbb{1}\{\mathcal{E}^c\}\right] + q\mathbb{E}\left[\frac{1}{\mathbb{E}_{\mathrm{Ber}\left(\frac{L+1}{n_{\mathrm{IS}}}\right)}[\mathrm{M}]} \cdot \mathbb{1}\{\mathcal{E}\}\right] \quad (4)$$

Assume for now that $q < p$ (we will later proof the opposite event), then $\frac{1}{\mathbb{E}_{\mathrm{Ber}\left(\frac{L+1}{n_{\mathrm{IS}}}\right)}[\mathrm{M}]}$ is strictly

non-increasing in L since $\frac{q}{p} < \frac{1-q}{1-p}$, and hence, when $\mathcal{E}^c$ holds and hence L concentration around the average that

$$\frac{1}{\mathbb{E}_{\mathrm{Ber}\left(\frac{L+1}{n_{\mathrm{IS}}}\right)}[\mathrm{M}]} \leq \frac{1}{\mathbb{E}_{\mathrm{Ber}\left(\frac{(L+1)\cdot(p-\eta_\delta)}{n_{\mathrm{IS}}}\right)}[\mathrm{M}]}$$

$$= \frac{1}{\frac{(n_{\mathrm{IS}}-1)(p-\eta_\delta)+1}{n_{\mathrm{IS}}}\frac{q}{p} + \frac{n_{\mathrm{IS}}-1-(n_{\mathrm{IS}}-1)(p-\eta_\delta)}{n_{\mathrm{IS}}}\frac{1-q}{1-p}}$$

$$= \frac{1}{\left(p - \frac{p}{n_{\mathrm{IS}}} + \frac{\eta_\delta}{n_{\mathrm{IS}}} - \eta_\delta + \frac{1}{n_{\mathrm{IS}}}\right)\frac{q}{p} + \left(1 - p - \frac{1}{n_{\mathrm{IS}}} + \frac{p}{n_{\mathrm{IS}}} + \eta_\delta - \frac{\eta_\delta}{n_{\mathrm{IS}}}\right)\frac{1-q}{1-p}}$$

$$= \frac{1}{1 + \left(\frac{q}{p} - \frac{1-q}{1-p}\right)\left(\frac{1-p+\eta_\delta-n\eta_\delta}{n_{\mathrm{IS}}}\right)}$$

$$= 1 + \sum_{\kappa=1}^{\infty}(-1)^\kappa \left(\frac{q}{p} - \frac{1-q}{1-p}\right)^\kappa \left(\frac{1-p+\eta_\delta-n\eta_\delta}{n_{\mathrm{IS}}}\right)^\kappa,$$

where the last step is by Taylor expansion. Using (4) and the monotonicity of $\frac{1}{\mathbb{E}_{\mathrm{Ber}\left(\frac{L+1}{n_{\mathrm{IS}}}\right)}[\mathrm{M}]}$, we write

$$\Pr(X_{\ell,I_\ell} = 1) = q\mathbb{E}\left[\frac{1}{\mathbb{E}_{\mathrm{Ber}\left(\frac{L+1}{n_{\mathrm{IS}}}\right)}[\mathrm{M}]}\right]$$

$$\leq q\left(1 + \sum_{\kappa=1}^{\infty}(-1)^\kappa \left(\frac{q}{p} - \frac{1-q}{1-p}\right)^\kappa \left(\frac{1-p+\eta_\delta-n\eta_\delta}{n_{\mathrm{IS}}}\right)^\kappa\right) + q\delta\frac{p}{q},$$

and hence

$$\Pr(X_{\ell,I_\ell} = 1) - q \leq \delta p + (1-\delta)\sum_{\kappa=1}^{\infty}(-1)^\kappa \left(\frac{q}{p} - \frac{1-q}{1-p}\right)^\kappa \left(\frac{1-p+\eta_\delta-n\eta_\delta}{n_{\mathrm{IS}}}\right)^\kappa$$

Similarly, we get by bounding $\frac{1}{\mathbb{E}_{\mathrm{Ber}\left(\frac{L+1}{n_{\mathrm{IS}}}\right)}[\mathrm{M}]} \geq \frac{1}{\mathbb{E}_{\mathrm{Ber}\left(\frac{(L+1)\cdot(p+\eta_\delta)}{n_{\mathrm{IS}}}\right)}[\mathrm{M}]}$ and using (4) that

$$\Pr(X_{\ell,I_\ell} = 1) - q \geq \delta q\frac{1-p}{1-q} + (1-\delta)\sum_{\kappa=1}^{\infty}(-1)^\kappa \left(\frac{q}{p} - \frac{1-q}{1-p}\right)^\kappa \left(\frac{1-p-\eta_\delta+n\eta_\delta}{n_{\mathrm{IS}}}\right)^\kappa \Leftrightarrow$$

$$q - \Pr(X_{\ell,I_\ell} = 1) \leq -\delta q\frac{1-p}{1-q} + (1-\delta)\sum_{\kappa=1}^{\infty}(-1)^{\kappa+1} \left(\frac{q}{p} - \frac{1-q}{1-p}\right)^\kappa \left(\frac{1-p-\eta_\delta+n\eta_\delta}{n_{\mathrm{IS}}}\right)^\kappa.$$

When $p \leq q$, then $\frac{1}{\mathbb{E}_{\mathrm{Ber}\left(\frac{L+1}{n_{\mathrm{IS}}}\right)}[\mathrm{M}]}$ is strictly non-decreasing, hence, under $\mathcal{E}$, we have

$$\frac{1}{\mathbb{E}_{\mathrm{Ber}\left(\frac{L+1}{n_{\mathrm{IS}}}\right)}[\mathrm{M}]} \leq \frac{1}{\mathbb{E}_{\mathrm{Ber}\left(\frac{(L+1)\cdot(p+\eta_\delta)}{n_{\mathrm{IS}}}\right)}[\mathrm{M}]} = 1 + \sum_{\kappa=1}^{\infty}(-1)^\kappa \left(\frac{q}{p} - \frac{1-q}{1-p}\right)^\kappa \left(\frac{1-p-\eta_\delta+n\eta_\delta}{n_{\mathrm{IS}}}\right)^\kappa,$$

and thus from (4) that

$$\Pr(X_{\ell,I_\ell} = 1) - q \le q\delta\frac{1-p}{1-q} + (1-\delta)\sum_{\kappa=1}^{\infty}(-1)^\kappa\left(\frac{q}{p} - \frac{1-q}{1-p}\right)^\kappa\left(\frac{1-p-\eta_\delta+n\eta_\delta}{n_{\text{IS}}}\right)^\kappa.$$

Similarly, we bound $\frac{1}{\mathbb{E}_{\text{Ber}\left(\frac{L+1}{n_{\text{IS}}}\right)}[M]} \le \frac{1}{\mathbb{E}_{\text{Ber}\left(\frac{(L+1)\cdot(p+\eta_\delta)}{n_{\text{IS}}}\right)}[M]}$ to obtain

$$\Pr(X_{\ell,I_\ell} = 1) - q \ge q\delta\frac{p}{q} + (1-\delta)\sum_{\kappa=1}^{\infty}(-1)^\kappa\left(\frac{q}{p} - \frac{1-q}{1-p}\right)^\kappa\left(\frac{1-p+\eta_\delta-n\eta_\delta}{n_{\text{IS}}}\right)^\kappa \Leftrightarrow$$

$$q - \Pr(X_{\ell,I_\ell} = 1) \le -q\delta\frac{p}{q} + (1-\delta)\sum_{\kappa=1}^{\infty}(-1)^{\kappa+1}\left(\frac{q}{p} - \frac{1-q}{1-p}\right)^\kappa\left(\frac{1-p+\eta_\delta-n\eta_\delta}{n_{\text{IS}}}\right)^\kappa$$

Since $0 \le p + \eta_\delta \le 1$ and $1 \ge p - \eta_\delta \ge 0$ by an appropriate choice of the concentration intervals, we have by approximations up to second order terms that

$$|\Pr(X_{\ell,I_\ell} = 1) - q| \le q\delta\max\left\{\frac{p}{q}, \frac{1-p}{1-q}\right\} + \eta_\delta\left(\frac{q}{p} - \frac{1-q}{1-p}\right) + \left(\frac{q}{p} - \frac{1-q}{1-p}\right)^2\mathcal{O}\left(\frac{1}{n_{\text{IS}}^2} + \eta_\delta^2\right)$$

$$= \frac{q}{n_{\text{IS}}^2}\left(\frac{p}{q} + \frac{1-p}{1-q}\right) + \mathcal{O}\left(\left[\left(\frac{q}{p} - \frac{1-q}{1-p}\right) + \left(\frac{q}{p} - \frac{1-q}{1-p}\right)^2\right]\sqrt{\frac{6p\log(2n_{\text{IS}})}{n_{\text{IS}}}}\right).$$

This concludes the proof. $\qquad\square$

*Proof of Lemma 1.* Using Lemma 2, we can show the following. Recall the following probability law of the stochastic quantizer $Q_s(\cdot)$ [Alistarh et al., 2017] using $s > 0$ quantization intervals, which takes as input the entry $x_e$ of a gradient $\mathbf{x} \in \mathbb{R}^d$ vector. Let $0 \le \tau_e < s$ be an integer such that $\frac{\tau_e}{s} \le \frac{|x_e|}{\|\mathbf{x}\|} \le \frac{\tau_e+1}{s}$, then $Q_s(x_e)$ is defined as $\text{Ber}\left(\frac{|x_e|}{\|\mathbf{x}\|}s - \tau_e\right)$, which outputs $\|\mathbf{x}\| \cdot \text{sign}(x_e)(\tau_e + 1)/s$ in case of success, and $\|\mathbf{x}\| \cdot \text{sign}(x_e)\tau_e/s$ otherwise.

Focusing on an entry $x_e$, we prove a contraction property for MRC with stochastic quantization with posterior $q_e = \frac{|x_e|}{\|\mathbf{x}\|}s - \tau_e$, and an arbitrary prior $p_e$. In fact, the MRC methodology $\mathcal{C}_{\text{mrc}}(\cdot)$ leads to sampling from an approximate distribution with parameter $\tilde{q}_e$. To be more specific, $\mathcal{C}_{\text{mrc}}(x_e)$ outputs $\|\mathbf{x}\| \cdot \text{sign}(x_e)(\tau_e + 1)/s$ with probability $\tilde{q}_e$, and $\|\mathbf{x}\| \cdot \text{sign}(x_e)\tau_e/s$ with probability $1 - \tilde{q}_e$. We established in Lemma 2 an upper bound on $|q_e - \tilde{q}_e|$, which will be useful in the following.

To prove a contraction property of the kind

$$\mathbb{E}[\|\mathcal{C}_{\text{mrc}}(\mathbf{x}) - \mathbf{x}\|_2^2] \le (1-\delta)\|\mathbf{x}\|^2,$$

we can write

$$\mathbb{E}[\|\mathcal{C}_{\text{mrc}}(\mathbf{x}) - \mathbf{x}\|^2] = \mathbb{E}\left[\sum_{e=1}^{d}(\mathcal{C}_{\text{mrc}}(x_e) - x_e)^2\right]$$

$$= \|\mathbf{x}\|^2\sum_{e=1}^{d}\mathbb{E}\left[\left(\frac{\mathcal{C}_{\text{mrc}}(x_e)}{\|\mathbf{x}\|} - \frac{x_e}{\|\mathbf{x}\|}\right)^2\right]$$

$$= \|\mathbf{x}\|^2\sum_{e=1}^{d}\left[\tilde{q}_e\left(\frac{\text{sign}(x_e)(\tau_e + 1)}{s} - \frac{x_e}{\|\mathbf{x}\|}\right)^2 + (1 - \tilde{q}_e)\left(\frac{\text{sign}(x_e)\tau_e}{s} - \frac{x_e}{\|\mathbf{x}\|}\right)^2\right]$$

$$= \|\mathbf{x}\|^2\sum_{e=1}^{d}\left[(\tilde{q}_e - q_e + q_e)\left(\frac{\tau_e + 1}{s} - \frac{|x_e|}{\|\mathbf{x}\|}\right)^2 + (1 - \tilde{q}_e - q_e + q_e)\left(\frac{\tau_e}{s} - \frac{|x_e|}{\|\mathbf{x}\|}\right)^2\right]$$

$$= \|\mathbf{x}\|^2\sum_{e=1}^{d}\left[(q_e + \tilde{q}_e - q_e)\left(\left(\frac{\tau_e}{s} - \frac{|x_e|}{\|\mathbf{x}\|}\right)^2 + \frac{1}{s^2} + \frac{1}{s}\left(\frac{\tau_e}{s} - \frac{|x_e|}{\|\mathbf{x}\|}\right)\right)\right.$$

$$+ (1 - q_e + q_e - \tilde{q}_e) \left( \frac{\tau_e}{s} - \frac{|x_e|}{\|\mathbf{x}\|} \right)^2 \Bigg]$$

$$= \|\mathbf{x}\|^2 \sum_{e=1}^d \left[ (\tilde{q}_e - q) \left( \frac{1}{s^2} + \frac{1}{s} \left( \frac{\tau_e}{s} - \frac{|x_e|}{\|\mathbf{x}\|} \right) \right) + q_e \left( \frac{1}{s^2} + \frac{1}{s} \left( \frac{\tau_e}{s} - \frac{|x_e|}{\|\mathbf{x}\|} \right) \right) + \left( \frac{\tau_e}{s} - \frac{|x_e|}{\|\mathbf{x}\|} \right)^2 \right],$$
$$\tag{5}$$

where

$$q_e \left( \frac{1}{s^2} + \frac{1}{s} \left( \frac{\tau_e}{s} - \frac{|x_e|}{\|\mathbf{x}\|} \right) \right)$$

$$= \left( \frac{|x_e|}{\|\mathbf{x}\|} s - \tau_e \right) \left( \frac{1}{s^2} + \frac{1}{s} \left( \frac{\tau_e}{s} - \frac{|x_e|}{\|\mathbf{x}\|} \right) \right)$$

$$= -s \left( \frac{\tau_e}{s} - \frac{|x_e|}{\|\mathbf{x}\|} \right) \frac{1}{s} \left( \frac{1}{s} + \left( \frac{\tau_e}{s} - \frac{|x_e|}{\|\mathbf{x}\|} \right) \right)$$

$$= - \left( \frac{\tau_e}{s} - \frac{|x_e|}{\|\mathbf{x}\|} \right)^2 - \frac{1}{s} \left( \frac{\tau_e}{s} - \frac{|x_e|}{\|\mathbf{x}\|} \right).$$

Substituting the result in (5), obtain

$$\mathbb{E}[\|\mathcal{C}_{\mathrm{mrc}}(\mathbf{x}) - \mathbf{x}\|\|^2] = \mathbb{E} \left[ \sum_{e=1}^d (\mathcal{C}_{\mathrm{mrc}}(x_e) - x_e)^2 \right]$$

$$= \|\mathbf{x}\|^2 \sum_{e=1}^d \left[ (\tilde{q}_e - q_e) \left( \frac{1}{s^2} + \frac{1}{s} \left( \frac{\tau_e}{s} - \frac{|x_e|}{\|\mathbf{x}\|} \right) \right) - \frac{1}{s} \left( \frac{\tau_e}{s} - \frac{|x_e|}{\|\mathbf{x}\|} \right) \right]$$

$$= \|\mathbf{x}\|^2 \sum_{e=1}^d \left[ (\tilde{q}_e - q_e) \frac{1}{s} \left( \frac{\tau_e + 1}{s} - \frac{|x_e|}{\|\mathbf{x}\|} \right) - \frac{1}{s} \left( \frac{\tau_e}{s} - \frac{|x_e|}{\|\mathbf{x}\|} \right) \right]$$

$$\leq \|\mathbf{x}\|^2 \sum_{e=1}^d \left[ |\tilde{q}_e - q_e| \frac{1}{s} \left( \frac{\tau_e + 1}{s} - \frac{|x_e|}{\|\mathbf{x}\|} \right) + \frac{1}{s} \left( \frac{|x_e|}{\|\mathbf{x}\|} - \frac{\tau_e}{s} \right) \right]$$

$$\leq \|\mathbf{x}\|^2 (|\tilde{q}_e - q_e| \frac{d}{s^2} + \frac{d}{s^2}),$$

where, by Lemma 2, we have for $\Delta_e := \frac{q_e}{p_e} - \frac{1-q_e}{1-p_e}$ and $\Delta'_e := q_e \left( \frac{p_e}{q_e} + \frac{1-p_e}{1-q_e} \right)$ that

$$|\tilde{q}_e - q_e| \leq \frac{\Delta'_e}{n_{\mathrm{IS}}^2} + \mathcal{O} \left( (\Delta_e + \Delta_e^2) \sqrt{\frac{6 p_e \log(2 n_{\mathrm{IS}})}{n_{\mathrm{IS}}}} \right).$$

Let $\bar{\Delta} := \max_{e \in [d]} \frac{q_e}{p_e} - \frac{1-q_e}{1-p_e}$, $\bar{\Delta}' := \max_{e \in [d]} q_e \left( \frac{p_e}{q_e} + \frac{1-p_e}{1-q_e} \right)$, and $\bar{p} := \max_{e \in [d]} p_e$. We will ensure that $\frac{\bar{\Delta}'}{n_{\mathrm{IS}}^2} + \mathcal{O} \left( (\bar{\Delta} + \bar{\Delta}^2) \sqrt{\frac{6 \bar{p} \log(2 n_{\mathrm{IS}})}{n_{\mathrm{IS}}}} \right) \leq 1$ by making each of the individual terms $\leq \frac{1}{2}$. By choosing $n_{\mathrm{IS}} \geq \sqrt{2 \bar{\Delta}'}$, we have $\frac{\bar{\Delta}'}{n_{\mathrm{IS}}^2} \leq \frac{1}{2}$. To ensure that $(\bar{\Delta} + \bar{\Delta}^2) \sqrt{\frac{6 \bar{p} \log(2 n_{\mathrm{IS}})}{n_{\mathrm{IS}}}} \leq \frac{1}{2}$, we require $\frac{\log(2 n_{\mathrm{IS}})}{n_{\mathrm{IS}}} \leq \frac{1}{\sqrt{6 \bar{p} (\bar{\Delta} + \bar{\Delta}^2)}}$. By Weinberger and Yemini [2023, Lemma 15], this holds when $n_{\mathrm{IS}} = \mathcal{O}(\log(6 \bar{p} (\bar{\Delta} + \bar{\Delta}^2)) \sqrt{6 \bar{p} (\bar{\Delta} + \bar{\Delta}^2)})$. Hence, choosing $n_{\mathrm{IS}} = \mathcal{O}(\max\{\sqrt{2 \bar{\Delta}'}, \log(6 \bar{p} (\bar{\Delta} + \bar{\Delta}^2)) \sqrt{6 \bar{p} (\bar{\Delta} + \bar{\Delta}^2)}\})$, we have $\frac{\bar{\Delta}'}{n_{\mathrm{IS}}^2} + \mathcal{O} \left( (\bar{\Delta} + \bar{\Delta}^2) \sqrt{\frac{6 \bar{p} \log(2 n_{\mathrm{IS}})}{n_{\mathrm{IS}}}} \right) \leq 1$. Thus, we have $0 \leq \delta \leq 1$ if $\frac{2d}{s^2} \leq 1$, and hence $s \geq \sqrt{2d}$. This concludes the proof.

□

*Proof of Theorem 1.* Assume a party estimates the Bernoulli distributions $Q_j$ with parameters $q_j$ held by parties $j \in [n]$. The estimating party shares with each of the other parties a common prior $P_j$ in the form of a Bernoulli distribution with parameter $p_j$ and access to unlimited shared randomness. To help estimate $Q_j$, the $j$-th party sends $K$ samples to the estimator through MRC. Therefore, both parties sample $Kn_{\mathrm{IS}}$ i.i.d. samples $X_{\ell,i} \sim P_j$ for $\ell \in [K], i \in [n_{\mathrm{IS}}]$, independently and identically from $P_j$. The party holding $Q_j$ constructs for each $\ell \in [K]$ an auxiliary distribution

$$W_\ell(i) = \frac{Q_j(X_{\ell,i})/P_j(X_{\ell,i})}{\sum_{i=1}^{n_{\mathrm{IS}}} Q_j(X_{\ell,i})/P_j(X_{\ell,i})},$$

from which it samples to obtain an index $I_\ell$. The index is transmitted to the estimating party, which reconstructs the corresponding sample $X_{\ell,I_\ell}$. Averaging the samples for all $\ell \in [K]$ gives an estimate $\hat{q}_j$ of $q_j$, i.e., $\hat{q}_j = \frac{1}{K} \sum_{\ell=1}^{K} X_{\ell,I_\ell}$. This process is repeated for all $j \in [n]$.

We assume that $|q_j - p_j| \le \rho$ for all $i, j \in [n]$, and that the difference between the priors, is bounded as $|p_i - p_j| \le \zeta$ for all $i, j \in [n]$. The goal is to bound $\mathrm{d}_{\mathrm{KL}}\left(\frac{1}{n} \sum_{j=1}^{n} \hat{q}_j \| p_i\right)$ from above for any $i \in [n]$.

By the convexity of KL-divergence, we have

$$\mathrm{d}_{\mathrm{KL}}\left(\frac{1}{n} \sum_{j=1}^{n} \hat{q}_j \| p_i\right) \le \frac{1}{n} \sum_{i=1}^{n} \mathrm{d}_{\mathrm{KL}}\left(\hat{q}_j \| p_i\right).$$

To bound $\mathrm{d}_{\mathrm{KL}}\left(\hat{q}_j \| p_i\right)$ for any $i, j \in [n]$, by the triangle inequality, we can write

$$|\hat{q}_j - p_i| \le |\hat{q}_j - \Pr(X_\ell = 1)| + |\Pr(X_\ell = 1) - q_j| + |q_j - p_j| + |p_j - p_i|,$$

where $|\hat{q}_j - \Pr(X_\ell = 1)|$ is bounded by Lemma 2. By Hoeffding's inequality, we have with probability at least $1 - \delta'$ that

$$|\hat{q} - \Pr(X_\ell = 1)| \le \sqrt{\frac{-\ln(\delta'/2)}{2n_{\mathrm{IS}}}}.$$

Thus, with probability at least $1 - \delta'$, since $p_j \le p_i + \zeta$, we have with $\Delta_j := \frac{q_j}{p_j - \zeta} - \frac{1-q_j}{1-p_j+\zeta}$ and $\Delta_j' := q_j\left(\frac{p_j+\zeta}{q_j} + \frac{1-p_j+\zeta}{1-q_j}\right)$ that

$$|\hat{q}_j - p_i| \le \frac{\Delta_j'}{n_{\mathrm{IS}}^2} + \mathcal{O}\left((\Delta_j + \Delta_j^2)\sqrt{\frac{6(p_i + \zeta)\log(2n_{\mathrm{IS}})}{n_{\mathrm{IS}}}}\right) + \sqrt{\frac{-\ln(\delta'/2)}{2n_{\mathrm{IS}}}} + \rho + \zeta.$$

This holds under the assumption that $p_j > \zeta$ for all $j \in [n]$. By the reversed Pinsker's inequality, we obtain

$$\mathrm{D}_{\mathrm{KL}}\left(\hat{q}_j \| p_i\right) \le \frac{2}{\min\{p_i, 1 - p_i\}} \left(\frac{\Delta_j'}{n_{\mathrm{IS}}^2} + \mathcal{O}\left((\Delta_j + \Delta_j^2)\sqrt{\frac{6(p_i + \zeta)\log(2n_{\mathrm{IS}})}{n_{\mathrm{IS}}}}\right)\right.$$
$$\left. + \sqrt{\frac{-\ln(\delta'/2)}{2n_{\mathrm{IS}}}} + \rho + \zeta\right)^2.$$

The statement of the theorem follows by the convexity of KL-divergence. $\square$

**Remark 1.** *Note that our analysis can be extended to other parametric distributions, such as multi-variate Gaussians. The key ingredient is to replace the specific upper bound on the bias in Lemma 2 with the generic results from Chatterjee and Diaconis [2018], which holds for all classes of distributions. Using this upper bound, one can follow our derivations to prove the communication costs in Theorem 1, i.e., using the convexity of KL divergence and decomposing the error in parameter estimation into a bias term and a concentration term to bound the sampling error. Standard concentration results can be utilized to bound the latter from above, e.g., Hoeffding's inequality for sub-Gaussian random variables. The remaining steps follow analogously to our proof. Similar adaptations apply to the contraction property in Lemma 1 necessary to establish convergence guarantees for conventional FL.*

# B    Convergence Analysis

Using the contraction property derived in Lemma 1, we can show that a straightforward extension of BICOMPFL-GR-CFL to error-feedback as used in [Richtárik et al., 2021] leads to the following convergence guarantee. The algorithmic details of the extension can be found in Algorithm 3. Therefore, assume that for all for $\mathbf{x}, \mathbf{y} \in \mathbb{R}^d$ and $i \in [n]$, the following Lipschitz property holds:

$$\|\nabla F(\mathbf{x}, \mathcal{D}_i) - \nabla F(\mathbf{y}, \mathcal{D}_i)\| \leq L_i \|\mathbf{x} - \mathbf{y}\|$$

Let $F(\theta) := \frac{1}{n} \sum_{i=1}^n \nabla F(\theta, \mathcal{D}_i)$ be the global loss function and $L' := \sqrt{\frac{1}{n} \sum_{i=1}^n L_i}$.

**Theorem 2.** *If $F^\star := \inf_{\theta \in \mathbb{R}^d} \{F(\theta)\} > -\infty$ and $\mathbb{E}[\|\mathbf{g}^t - \nabla F(\theta_t)\|^2] \leq \sigma^2$, then with $\eta \leq \left(L + L' \sqrt{\frac{1-\delta}{(1-\sqrt{1-\delta})^2}}\right)^{-1}, L = 1, s \geq \sqrt{2d}$, and $n_{IS}$ satisfying Lemma 1 in every iteration $t$, we have for Algorithm 3 that*

$$\sum_{t=1}^T \mathbb{E}\left[\|F(\theta_t)\|^2\right] \leq \frac{2(F(\theta_0) - F^\star)}{\eta T} + \frac{\sigma^2}{(1 - \sqrt{1-\delta})T}.$$

Similarly, guarantees can be derived for other algorithms, such as modified versions of BICOMPFL-PR with error-feedback and momentum, using Lemma 1. However, we emphasize the generality of BICOMPFL, reaching beyond conventional FL with stochastic compression to pure stochastic narratives.

**Remark 2.** *The choice of $s$ depends on the model architecture and dataset complexities, affecting the number of iterations required until convergence. The simpler the learning task, the fewer quantization intervals are sufficient for convergence. Beyond generic arguments, we particularly highlight that $s$ should be carefully selected, respecting the expected variance of the gradients. If gradients exhibit inherently large variance, e.g., through small mini-batch sizes, large variance of the local datasets, or substantially over-parameterized networks, fewer quantization intervals might be efficient, and increasing that number would only negligibly improve the performance. We also note that one-bit quantization has been shown to be often remarkably effective despite its simplicity [Seide et al., 2014, Karimireddy et al., 2019].*

# C    Gradient Descent with a KL-Proximity

Mirror descent employs point-wise optimization in the form of a first-order approximation of $F(\hat{\theta}_t, \mathcal{D}_i)$ with proximity term $D_F(p, q)$, where $D_F$ is the Bregman divergence associated with function $F(\cdot)$. When $F(x) = \|x\|^2$, and hence the Bregman divergence is the Euclidean distance, this is known as gradient descent. Let now $p$ and $q$ be vectors with the entries corresponding to independent Bernoulli parameters. When we choose $F(x) = x \log(x) + (1-x) \log(1-x)$, the Bregman divergence becomes $D_F(p, q) = \sum_{k=1}^d D_{\text{KL}}(p_k \| q_k)$. Hence, we are optimizing with respect to a KL-proximity constraint. The mapping between dual and primal spaces is then given by $\nabla F(x) = \log(x) - \log(1-x)$ and $(\nabla F(x))^{-1} = \frac{1}{e^{-x}+1}$, respectively; also known as the inverse sigmoid and the sigmoid functions.

# D    Block Allocation

The simplest yet effective strategy for block allocation is to partition the model into equally-sized blocks of size $d/B$ for MRC (Fixed). The partitioning into blocks is required to make MRC practically feasible in this setting. It is known that for vanishing MRC error, the number of samples $n_{\text{IS}}$ from a block $p_{i,u,b}^t$ of the prior is supposed to be in the order of $\exp\left(D_{\text{KL}}\left(q_{i,b}^t \| p_{i,u,b}^t\right)\right)$, where $q_{i,b}^t$ is the $b$-th block of posterior $q_i^t$. It was observed by Isik et al. [2024] that the KL-divergence decreases as the training progresses with the global model used as a prior, which is intuitive since the local training will change the posterior less and less as training converges. To adapt the block size according to the

---

**Algorithm 3** BICOMPFL-GR-CFL with stochastic quantization $Q_s(\cdot)$ and EF21 from Richtárik et al. [2021]

---

**Require:** Both clients and federator initialize the same global model $\theta_0$ using a shared seed

**Ensure:** Set $t = 0$, clients set prior $p^t = \hat{\theta}_0 = \theta_0, \forall i \in [n]$, clients compute and broadcast $\mathbf{v}_i^0 = \mathcal{C}_{\mathrm{mrc}}(Q_s(g_i^t), p^t)$, with $g_i^t$ the local gradient for $\theta_0$; hence, $\mathbf{v}^0 = \frac{1}{n}\sum_{i=1}^n \mathbf{v}_i^0$ public

1: Update $\forall i : \hat{\theta}_{t+1} = \hat{\theta}_t - \eta\mathbf{v}^{t+1}$
2: **repeat**
3:   **for** Client $i \in [n]$ **do**
4:     Compute gradient $g_i^t$ by local training over $L$ local iterations
5:     Stochastic compression $q_i^t \leftarrow Q_s(g_i^t - \mathbf{v}_i^t)$
6:     Sample indices $I_{i,\ell}^b, \ell \in [n_{\mathrm{UL}}], b \in [B]$ from $q_i^t$ with prior $p^t$ and transmit to federator to reconstruct $\hat{q}_i^t = \mathcal{C}_{\mathrm{mrc}}(q_i^t, p^t)$
7:     Update $\mathbf{v}_i^{t+1} = \mathbf{v}_i^t + \hat{q}_i^t$
8:   **end for**
9:   Federator reconstructs and computes $\mathbf{v}^{t+1} = \mathbf{v}^t + \frac{1}{n}\sum_{j=1}^n \hat{q}_j^t$ from $\{I_{i,\ell}^b\}$
10:  Federator updates $\theta_{t+1} = \theta_t - \eta\mathbf{v}^{t+1}$
11:  Federator relays to client $j$ the other clients' indices $\{I_{i,\ell}^b\}_{\ell\in[n_{\mathrm{UL}}],b\in[B],i\in[n]\setminus\{j\}}$
12:  **for** Clients $i \in [n]$ **do**
13:    Reconstruct and compute $\mathbf{v}^{t+1} = \mathbf{v}^t + \frac{1}{n}\sum_{j=1}^n \hat{q}_j^t$ from $\{I_{i,\ell}^b\}$
14:    Update $\hat{\theta}_{t+1} = \hat{\theta}_t - \eta\mathbf{v}^{t+1}$ from $\{I_{i,\ell}^b\}$
15:  **end for**
16:  Clients and federator set prior $p^t = \hat{\theta}_{t+1}$
17:  $t \leftarrow t + 1$
18: **until** Convergence

---

divergence from the posterior with respect to the prior, Isik et al. [2024] proposed an adaptive block allocation strategy (Adaptive), where upon realizing a large deviation from the target KL-divergence per block, clients partition their model into blocks with equal sums of parameter-wise KL-divergences and transmit the block intervals to the federator. The federator aggregates the indices of all the clients, and broadcasts the updated block allocation.

We propose in this work a low complexity solution that adapts the block size according to the average KL-divergence per block (Adaptive-Avg). This alleviates the cost of computing and transmitting the exact block partitions, where the transmission of each block size requires $\log_2(b_{\max})$ bits, with $b_{\max}$ the maximum pre-defined block size. Instead, the transmission of one size is enough in our solution. If the average KL per block $\mathrm{D}_{\mathrm{KL}}\left(q_{i,b}^t \| p_{i,u,b}^t\right)$ deviates more than a given factor, the clients request to update the blocks. In the next iteration, each client proposes a block size, and the federator averages and broadcasts an updated size.

# E   Additional Experimental Details

We use the cross-entropy loss and a batch size of 128 in all our experiments. We use Adam [Kingma and Ba, 2015] as an optimizer with learning rate $\eta = 0.0003$ for all non-stochastic methods, and $\eta = 0.1$ for probabilistic mask training. For non-stochastic FL, we use a federator (server) learning rate of $0.1$, i.e., the clients' gradients are averaged, and the federator updates the global model with learning rate $0.1$, and with a learning rate of $0.005$ for BICOMPFL-GR with SignSGD. For M3, we use a federator learning rate of $0.02$ to obtain reliable results. For LIEC and CSER, we use an average period of 50 global iterations (cf. [Cheng et al., 2024, Xie et al., 2020]). For M3, we use TopK with $K = \lfloor d/n \rfloor$. To run the simulations, we use a cluster of different architectures, which we list in the following table.

Table 1: System specifications of our simulation cluster.

| CPU(s) | RAM | GPU(s) | VRAM |
|---|---|---|---|
| 2x Intel Xeon Platinum 8176 (56 cores) | 256 GB | 2x NVIDIA GeForce GTX 1080 Ti | 11 GB |
| 2x AMD EPYC 7282 (32 cores) | 512 GB | NVIDIA GeForce RTX 4090 | 24 GB |
| 2x AMD EPYC 7282 (32 cores) | 640 GB | NVIDIA GeForce RTX 4090 | 24 GB |
| 2x AMD EPYC 7282 (32 cores) | 448 GB | NVIDIA GeForce RTX 4080 | 16 GB |
| 2x AMD EPYC 7282 (32 cores) | 256 GB | NVIDIA GeForce RTX 4080 | 16 GB |
| HGX-A100 (96 cores) | 1 TB | 4x NVIDIA A100 | 80 GB |
| DGX-A100 (252 cores) | 2 TB | 8x NVIDIA Tesla A100 | 80 GB |
| DGX-1-V100 (76 cores) | 512 GB | 8x NVIDIA Tesla V100 | 16 GB |
| DGX-1-P100 (76 cores) | 512 GB | 8x NVIDIA Tesla P100 | 16 GB |
| HPE-P100 (28 cores) | 256 GB | 4x NVIDIA Tesla P100 | 16 GB |

The details of the CNN architectures used in our experiments are summarized in the following. The parameter count is $61706$ for LeNet5, $1933258$ for 4CNN, and $2262602$ for 6CNN.

Table 2: LeNet5 Architecture Overview

| Layer | Specification | Activation |
|---|---|---|
| 5x5 Conv | 6 filters, stride 1 | ReLU, AvgPool (2x2) |
| 5x5 Conv | 16 filters, stride 1 | ReLU, AvgPool (2x2) |
| Linear | 120 units | ReLU |
| Linear | 84 units | ReLU |
| Linear | 10 units | Softmax |

Table 3: 4-layer CNN (4CNN) Architecture Overview

| Layer | Specification | Activation |
|---|---|---|
| 3x3 Conv | 64 filters, stride 1 | ReLU |
| 3x3 Conv | 64 filters, stride 1 | ReLU, MaxPool (2x2) |
| 3x3 Conv | 128 filters, stride 1 | ReLU |
| 3x3 Conv | 128 filters, stride 1 | ReLU, MaxPool (2x2) |
| Linear | 256 units | ReLU |
| Linear | 256 units | ReLU |
| Linear | 10 units | Softmax |

Table 4: 6-layer CNN (6CNN) Architecture Overview

| Layer | Specification | Activation |
|---|---|---|
| 3x3 Conv | 64 filters, stride 1 | ReLU |
| 3x3 Conv | 64 filters, stride 1 | ReLU, MaxPool (2x2) |
| 3x3 Conv | 128 filters, stride 1 | ReLU |
| 3x3 Conv | 128 filters, stride 1 | ReLU, MaxPool (2x2) |
| 3x3 Conv | 256 filters, stride 1 | ReLU |
| 3x3 Conv | 256 filters, stride 1 | ReLU, MaxPool (2x2) |
| Linear | 256 units | ReLU |
| Linear | 256 units | ReLU |
| Linear | 10 units | Softmax |

For the sake of clarity, in the paper we restrict the analysis to a fixed number of importance samples $n_{\text{IS}}$, block sizes $B$, and choice of priors $p_{i,u}^t, p_{i,d}^t$. Our experiments have shown that, while increasing $n_{\text{IS}}$ beyond the ones used in our algorithms slightly improves the convergence over the number of epochs, the convergence with respect to the communication cost did not significantly improve. The block size is mainly limited by the system resources at hand, and one would choose the largest possible for best efficiency while complying with memory resources. We investigated many different

---

**Algorithm 4** Local Training at Client $i$

---

**Require:** Model $\hat{\theta}_{i,t}$
1: Map model to scores in the dual space: $\mathbf{s}_{i,t}^{(0)} = \sigma^{-1}(\hat{\theta}_{i,t}) = \log\left(\frac{\hat{\theta}_{i,t}}{1-\hat{\theta}_{i,t}}\right)$
2: **for** Local iterations $m \in [L]$ **do**
3:   $\mathbf{s}_{i,t}^{(m)} = \mathbf{s}_{i,t}^{(m-1)} - \eta \nabla_{\mathbf{s}_{i,t}^{(m-1)}} F(\hat{\theta}_{i,t}^{(m-1)}, \mathcal{D}_i)$, where $\hat{\theta}_{i,t}^{(m-1)} = \sigma(\mathbf{s}_{i,t}^{(m-1)})$
4: **end for**
5: Map back to primal space: $q_i^t = \sigma(\mathbf{s}_{i,t}^{(L)})$

---

prior choices and found the former global model to be reasonably good in almost all cases. With high heterogeneity, it might be beneficial to use different convex combinations as priors, which mix the former global model with the latest posterior estimate of a certain client, but the gains we experienced were minor. Hence, we settled on the former global estimate for simplicity in presenting the algorithm.

## F   Federated Probabilistic Mask Training

The idea in federated probabilistic mask training (FedPM) Isik et al. [2023] is to collaboratively train a probabilistic mask that determines which weights to maintain from a randomly initialized network. The motivation stems from the *lottery-ticket hypothesis* [Frankle and Carbin, 2019], which claims that randomly initialized networks contain sub-networks capable of reaching accuracy comparable to that of the full network. The weights $w$ of the network are randomly initialized at the start of training, and remain fixed. The federator and clients only train a mask, which determines for each parameter whether it is activated or not, i.e., identifying an efficient subnetwork within the given fixed network. The probabilistic masks $\theta_t$ are described by Bernoulli distributions, i.e., $\theta_t \in [0, 1]^d$ contains a Bernoulli parameter to be trained for each weight of the network. These parameters determine the probability of retaining the corresponding weights. During inference, the weights $w$ are masked with samples $x^t \in \{0, 1\}^d \sim \theta_t$ from the distribution $\theta_t$, i.e., the inference is conducted on a network with weights $w \odot x^t$. In FedPM, clients sample from their locally trained models, and send these samples to the federator, which, in turn, updates the global model by averaging these samples. The communication cost of this scheme is fixed for all iterations, even though the communication cost can be reduced since the KL-divergence between the global model and the locally trained models diminishes as the training progresses.

We adopt the following federated learning procedure for collaboratively learning network masks, and highlight in the following the parallels to mirror descent by referring to primal and dual spaces. Starting from a common model $\theta_0$, at iteration $t$, each client $i$ locally trains the model $\hat{\theta}_{i,t}$ in $L$ local iterations. To enable gradient descent, the model $\hat{\theta}_{i,t}$ is mapped to scores $\mathbf{s}_{i,t}^{(0)}$ in a dual space by the inverse Sigmoid function $\mathbf{s}_{i,t}^{(0)} = \sigma^{-1}(\hat{\theta}_{i,t}) = \log(\hat{\theta}_{i,t}) - \log(1 - \hat{\theta}_{i,t})$. The scores are then trained for $L$ local iterations $m \in [L]$ by computing the gradient $\nabla_{\mathbf{s}_{i,t}^{(\ell-1)}} F(\hat{\theta}_{i,t}^{(m-1)}, \mathcal{D}_i)$, where the straight-through estimator is used to compute the gradient of the non-differentiable Bernoulli sampling operation based on the distribution $\hat{\theta}_{i,t}^{(m-1)} = \sigma(\mathbf{s}_{i,t}^{(\ell-1)})$, i.e., the gradient equals the Bernoulli parameter. By mapping the model back to the primal space, each client $i$ obtains a model update in terms of a posterior $q_i^t = \sigma(\mathbf{s}_{i,t}^{(L)})$. The client training process is summarized in Algorithm 4.

## G   Minimal Random Coding (MRC)

Isik et al. [2024] proposed a method, called KL minimization with side information (KLMS), to reduce the cost of transmitting the local models $q_i^t$ to the federator. Consequently, the communication cost depends on the KL-divergence between the desired distribution and the common prior. This method utilizes the common side information available at both the clients and the federator, as well as shared randomness. The idea is that instead of sampling locally and sending the samples to the

federator, the federator in the KLMS method samples from the desired distribution through MRC. In a nutshell, MRC [Havasi et al., 2019] is based on importance sampling [Srinivasan, 2002] and makes use of a common prior to sample from a desired distribution. Consider two distributions $P$ and $Q$, where $P$ is known to both parties, and $Q$ is only known to the client. To make the federator sample from $Q$, both parties sample $n_{\text{IS}}$ samples $\{X_i\}_{i \in [n_{\text{IS}}]}$ from $P$. The client forms an auxiliary distribution $W(i) = \frac{Q(X_i)/P(X_i)}{\sum_{i=1}^{n_{\text{IS}}} Q(X_i)/P(X_i)}$ capturing the importance of the samples. A sample from $W$ is fully described by its index $i$, which can be transmitted with $\log_2(n_{\text{IS}})$ bits, and approximates a sample from $Q$. Chatterjee and Diaconis [2018] shown that importance sampling with posterior $Q$ and prior $P$ requires $n_{\text{IS}}$ to be in the order of $\Theta(\exp(\text{D}_{\text{KL}}(Q\|P)))$, where $\text{D}_{\text{KL}}(Q\|P)$ denotes the KL-divergence between distributions $Q$ and $P$. In what follows, we will also denote the KL-divergence between two Bernoulli distributions $Q$ and $P$ with parameters $q$ and $p$ by $\text{d}_{\text{KL}}(q\|p)$.

# H   Additional Experiments

We provide in the following experiments for both uniform (i.i.d.) and heterogeneous (non-i.i.d.) data distributions for training LeNet5 and a 4-layer CNN on MNIST, a 4-layer CNN on Fashion MNIST, and a 6-layer CNN on CIFAR-10. The details of the neural networks can be found in Tables 2 to 4. For each setting and method depicted, we show the average of three simulation runs with different seeds. We plot for each setting the test accuracies over the communication cost in bits, and the maximum test accuracy over the bitrate. We provide tables summarizing the maximum test accuracies with their standard deviation over multiple runs, the total bitrates and the bitrates split into uplink and downlink. The overall bitrates per parameter (bpp) are computed assuming point-to-point links between all participants, i.e., uplink and downlink costs have equal weight. For the case when a broadcast (BC) link between the federator and the clients is available, the bitrate per parameter for all baseline schemes reduces by a factor of $n$. BICOMPFL-GRprofits similarly from the broadcast link, but BICOMPFL-PRcannot profit due to the absence of shared randomness, giving the same overall bitrate compared to the point-to-point link scenario. We highlight for each of the measures the scheme with the best result. Consistently throughout all experiments, BICOMPFL achieves order-wise savings in the bitrates per parameter while reaching state-of-the-art accuracies in the classification task. While the sampling can introduce an additional computational overhead depending on the implementation, the storage cost is similar to the baselines. Since we leverage as priors the former global model, the additional storage cost incurred is limited to storing until the next iteration the estimate of the former global model at each client, i.e., where the training started, which is usually not a bottleneck. This can be cheaper than some baselines, which require storing data for momentum and error-feedback.

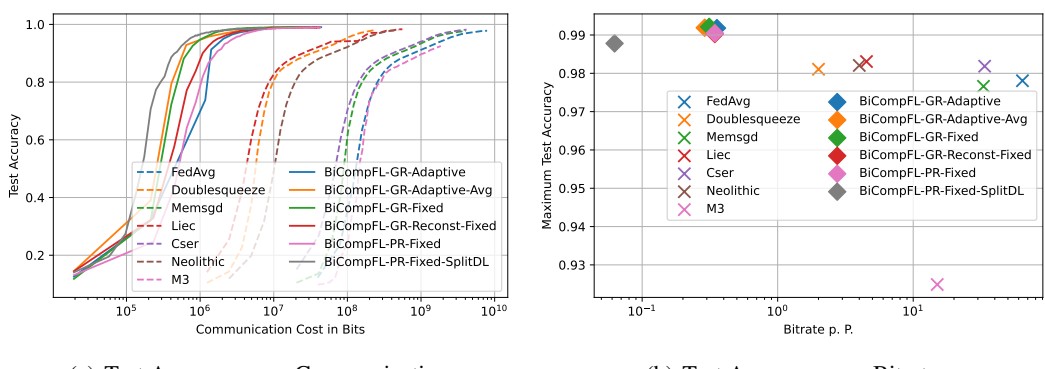

(a) Test Accuracy over Communication
(b) Test Accuracy over Bitrate

Figure 3: MNIST LeNet i.i.d.

For LeNet5 on MNIST, it can be observed that all our proposed methods converge significantly faster to satisfying accuracies with respect to the communication cost, while achieving higher maximum accuracies after 200 epochs than the non-stochastic baselines. Partitioning the model on the downlink can help to further reduce the communication cost with only a minor loss in performance, especially in the i.i.d. setting. For non-i.i.d. data distribution, the loss in performance is larger than for i.i.d.

distribution. However, at the beginning of the training, the model improves faster with respect to the communication cost than all other schemes. The bitrates are comparable for all our methods, with the exception of BiCompFL-PR-Fixed-SplitDL. Further, BiCompFL-GR-Reconst-Fixed does not suffer notable performance degradation from employing an additional MRC step (especially for i.i.d. data allocation).

Table 5: MNIST LeNet i.i.d.

| Method | Acc (mean ± std) | bpp | bpp (BC) | Uplink | Downlink |
|---|---|---|---|---|---|
| FedAvg | 0.978 ± 0.1 | 64.0 | 35.0 | 32.0 | 32.0 |
| Doublesqueeze | 0.981 ± 0.1 | 2.0 | 1.1 | 1.0 | 1.0 |
| Memsgd | 0.977 ± 0.1 | 33.0 | 4.2 | 1.0 | 32.0 |
| Liec | 0.983 ± 0.1 | 4.5 | 2.5 | 2.3 | 2.3 |
| Cser | 0.982 ± 0.09 | 34.0 | 4.3 | 1.0 | 33.0 |
| Neolithic | 0.982 ± 0.1 | 4.0 | 2.2 | 2.0 | 2.0 |
| M3 | 0.925 ± 0.2 | 15.0 | 2.2 | 8.0 | 7.1 |
| BiCompFL-GR-Adaptive | **0.992 ± 0.0006** | 0.36 | 0.068 | 0.036 | 0.32 |
| BiCompFL-GR-Adaptive-Avg | 0.992 ± 0.0003 | 0.29 | **0.055** | **0.029** | 0.26 |
| BiCompFL-GR-Fixed | 0.992 ± 0.0002 | 0.31 | 0.059 | 0.031 | 0.28 |
| BiCompFL-GR-Reconst-Fixed | 0.99 ± 0.0002 | 0.34 | 0.063 | 0.031 | 0.31 |
| BiCompFL-PR-Fixed | 0.99 ± 0.0004 | 0.34 | 0.34 | 0.031 | 0.31 |
| BiCompFL-PR-Fixed-SplitDL | 0.988 ± 0.0009 | **0.063** | 0.063 | 0.031 | **0.031** |

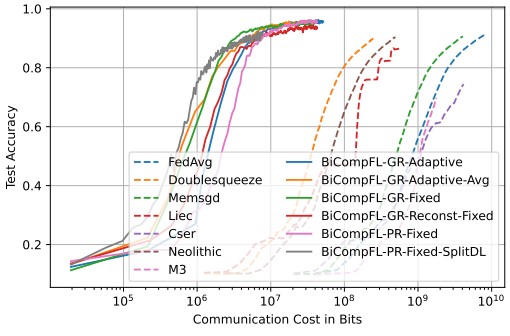

(a) Test Accuracy over Communication

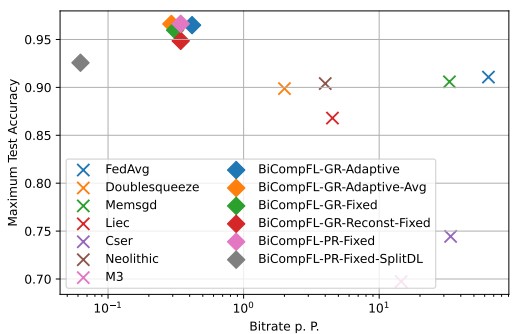

(b) Test Accuracy over Bitrate

Figure 4: MNIST LeNet non-i.i.d.

Table 6: MNIST LeNet non-i.i.d.

| Method | Acc (mean ± std) | bpp | bpp (BC) | Uplink | Downlink |
|---|---|---|---|---|---|
| FedAvg | 0.911 ± 0.2 | 64.0 | 35.0 | 32.0 | 32.0 |
| Doublesqueeze | 0.899 ± 0.2 | 2.0 | 1.1 | 1.0 | 1.0 |
| Memsgd | 0.906 ± 0.2 | 33.0 | 4.2 | 1.0 | 32.0 |
| Liec | 0.866 ± 0.2 | 4.5 | 2.5 | 2.3 | 2.3 |
| Cser | 0.744 ± 0.2 | 34.0 | 4.3 | 1.0 | 33.0 |
| Neolithic | 0.904 ± 0.2 | 4.0 | 2.2 | 2.0 | 2.0 |
| M3 | 0.697 ± 0.2 | 15.0 | 2.2 | 7.3 | 7.2 |
| BiCompFL-GR-Adaptive | 0.965 ± 0.02 | 0.42 | 0.079 | 0.042 | 0.37 |
| BiCompFL-GR-Adaptive-Avg | **0.966 ± 0.02** | 0.29 | **0.056** | **0.029** | 0.26 |
| BiCompFL-GR-Fixed | 0.96 ± 0.03 | 0.31 | 0.059 | 0.031 | 0.28 |
| BiCompFL-GR-Reconst-Fixed | 0.949 ± 0.03 | 0.34 | 0.063 | 0.031 | 0.31 |
| BiCompFL-PR-Fixed | 0.966 ± 0.02 | 0.34 | 0.34 | 0.031 | 0.31 |
| BiCompFL-PR-Fixed-SplitDL | 0.926 ± 0.04 | **0.063** | 0.063 | 0.031 | **0.031** |

For 4CNN trained on MNIST, the differences between the proposed approaches become more visible. In the i.i.d. setting, we can observe that the adaptive block allocations (both Adaptive and Adaptive-Avg) can drastically reduce the average bitrate in BICOMPFL-GR. Partitioning the model in the downlink (BICOMPFL-PR-Fixed-SplitDL) improves the accuracy over bitrate significantly compared to BICOMPFL-PR-Fixed.

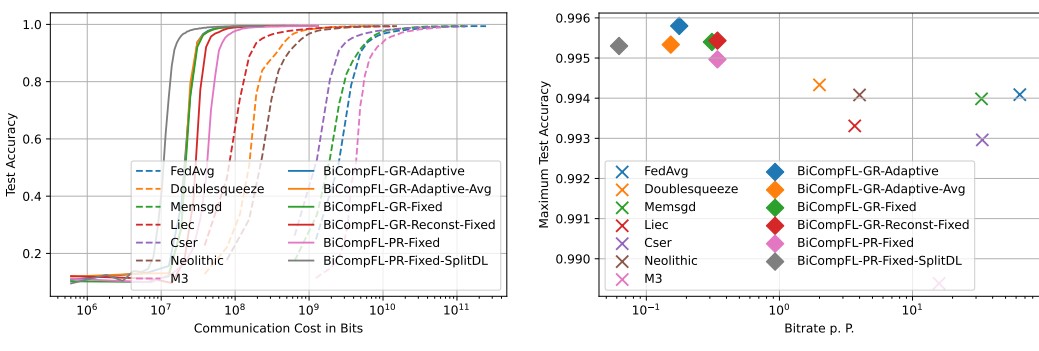

(a) Test Accuracy over Communication

(b) Test Accuracy over Bitrate

Figure 5: MNIST 4CNN i.i.d.

Table 7: MNIST 4CNN i.i.d.

| Method | Acc (mean ± std) | bpp | bpp (BC) | Uplink | Downlink |
|---|---|---|---|---|---|
| FedAvg | 0.994 ± 0.06 | 64.0 | 35.0 | 32.0 | 32.0 |
| Doublesqueeze | 0.994 ± 0.1 | 2.0 | 1.1 | 1.0 | 1.0 |
| Memsgd | 0.994 ± 0.08 | 33.0 | 4.2 | 1.0 | 32.0 |
| Liec | 0.993 ± 0.07 | 3.7 | 2.0 | 1.8 | 1.8 |
| Cser | 0.993 ± 0.06 | 33.0 | 4.3 | 1.0 | 32.0 |
| Neolithic | 0.994 ± 0.08 | 4.0 | 2.2 | 2.0 | 2.0 |
| M3 | 0.989 ± 0.2 | 16.0 | 2.2 | 8.4 | 7.4 |
| BiCompFL-GR-Adaptive | **0.996 ± 0.0001** | 0.18 | 0.034 | 0.018 | 0.16 |
| BiCompFL-GR-Adaptive-Avg | 0.995 ± 0.0001 | 0.15 | **0.029** | **0.015** | 0.14 |
| BiCompFL-GR-Fixed | 0.995 ± 0.0002 | 0.31 | 0.059 | 0.031 | 0.28 |
| BiCompFL-GR-Reconst-Fixed | 0.995 ± 0.0001 | 0.34 | 0.062 | 0.031 | 0.31 |
| BiCompFL-PR-Fixed | 0.995 ± 0.0002 | 0.34 | 0.34 | 0.031 | 0.31 |
| BiCompFL-PR-Fixed-SplitDL | 0.995 ± 0.0002 | **0.062** | 0.062 | 0.031 | **0.031** |

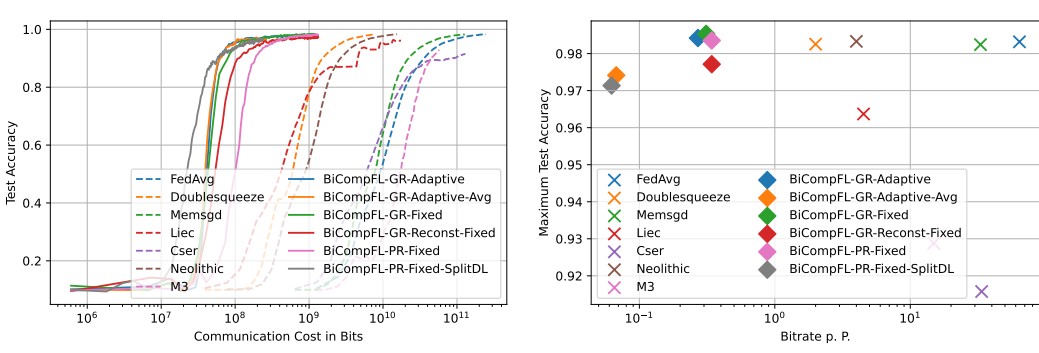

(a) Test Accuracy over Communication

(b) Test Accuracy over Bitrate

Figure 6: MNIST 4CNN non-i.i.d.

In the non-i.i.d. case of 4CNN on MNIST, the adaptive average allocation strategy provides a significant reduction in the bitrate for BICOMPFL-GR, with similar loss in the accuracy as SplitDL for BICOMPFL-PR. In this setting, it is also apparent that the reconstruction in BICOMPFL-GR degrades the performance without gains in the bitrate compared to the proposed Algorithm 1.

Table 8: MNIST 4CNN non-i.i.d.

| Method | Acc (mean ± std) | bpp | bpp (BC) | Uplink | Downlink |
|---|---|---|---|---|---|
| FedAvg | 0.983 ± 0.1 | 64.0 | 35.0 | 32.0 | 32.0 |
| Doublesqueeze | 0.982 ± 0.2 | 2.0 | 1.1 | 1.0 | 1.0 |
| Memsgd | 0.982 ± 0.2 | 33.0 | 4.2 | 1.0 | 32.0 |
| Liec | 0.963 ± 0.2 | 4.5 | 2.5 | 2.3 | 2.3 |
| Cser | 0.915 ± 0.1 | 34.0 | 4.3 | 1.0 | 33.0 |
| Neolithic | 0.983 ± 0.2 | 4.0 | 2.2 | 2.0 | 2.0 |
| M3 | 0.929 ± 0.3 | 15.0 | 2.2 | 7.8 | 7.1 |
| BiCompFL-GR-Adaptive | 0.984 ± 0.009 | 0.27 | 0.051 | 0.026 | 0.24 |
| BiCompFL-GR-Adaptive-Avg | 0.974 ± 0.02 | 0.067 | **0.013** | **0.0068** | 0.061 |
| BiCompFL-GR-Fixed | **0.985 ± 0.008** | 0.31 | 0.059 | 0.031 | 0.28 |
| BiCompFL-GR-Reconst-Fixed | 0.977 ± 0.01 | 0.34 | 0.062 | 0.031 | 0.31 |
| BiCompFL-PR-Fixed | 0.984 ± 0.009 | 0.34 | 0.34 | 0.031 | 0.31 |
| BiCompFL-PR-Fixed-SplitDL | 0.971 ± 0.02 | **0.062** | 0.062 | 0.031 | **0.031** |

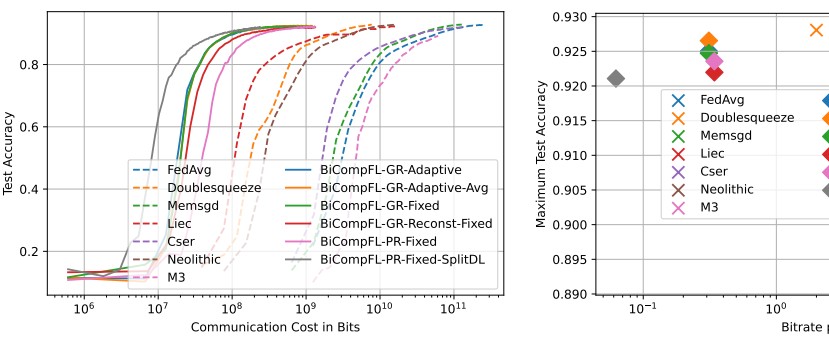

(a) Test Accuracy over Communication

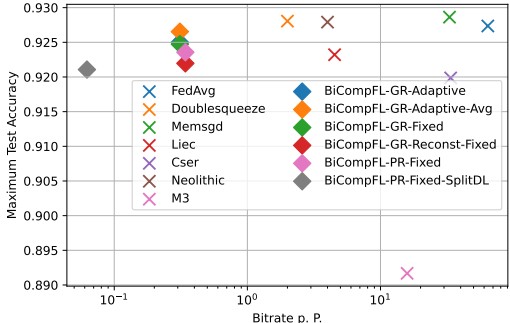

(b) Test Accuracy over Bitrate

Figure 7: Fashion MNIST 4CNN i.i.d.

Table 9: Fashion MNIST 4CNN i.i.d.

| Method | Acc (mean ± std) | bpp | bpp (BC) | Uplink | Downlink |
|---|---|---|---|---|---|
| FedAvg | 0.927 ± 0.07 | 64.0 | 35.0 | 32.0 | 32.0 |
| Doublesqueeze | **0.928 ± 0.1** | 2.0 | 1.1 | 1.0 | 1.0 |
| Memsgd | 0.928 ± 0.09 | 33.0 | 4.2 | 1.0 | 32.0 |
| Liec | 0.923 ± 0.08 | 4.5 | 2.5 | 2.3 | 2.3 |
| Cser | 0.92 ± 0.08 | 34.0 | 4.3 | 1.0 | 33.0 |
| Neolithic | 0.928 ± 0.09 | 4.0 | 2.2 | 2.0 | 2.0 |
| M3 | 0.892 ± 0.2 | 16.0 | 2.2 | 8.3 | 7.6 |
| BiCompFL-GR-Adaptive | 0.925 ± 0.001 | 0.31 | **0.059** | **0.031** | 0.28 |
| BiCompFL-GR-Adaptive-Avg | 0.927 ± 0.0007 | 0.31 | **0.059** | **0.031** | 0.28 |
| BiCompFL-GR-Fixed | 0.925 ± 0.0007 | 0.31 | **0.059** | **0.031** | 0.28 |
| BiCompFL-GR-Reconst-Fixed | 0.922 ± 0.001 | 0.34 | 0.062 | **0.031** | 0.31 |
| BiCompFL-PR-Fixed | 0.924 ± 0.002 | 0.34 | 0.34 | **0.031** | 0.31 |
| BiCompFL-PR-Fixed-SplitDL | 0.921 ± 0.002 | **0.062** | 0.062 | **0.031** | **0.031** |

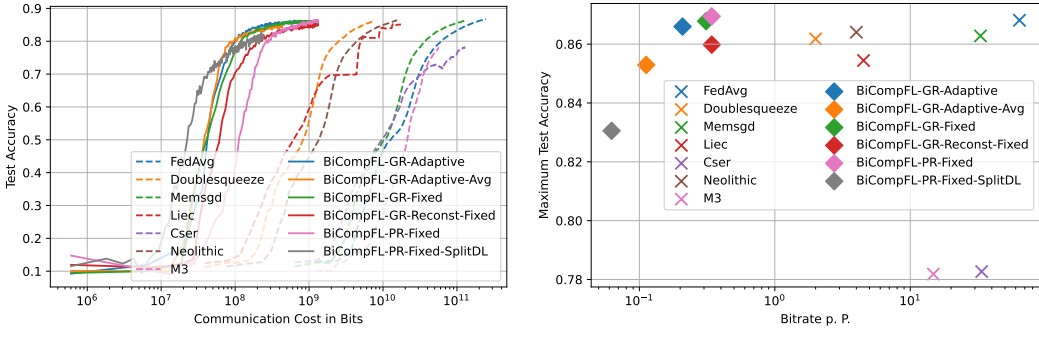

(a) Test Accuracy over Communication

(b) Test Accuracy over Bitrate

Figure 8: Fashion MNIST 4CNN non-i.i.d.

The results for Fashion MNIST are similar compared to the MNIST case. However, it becomes clear that BICOMPFL-PR can significantly suffer from the unavailability of shared randomness in terms of the achieved accuracy when data is highly heterogeneous.

Table 10: Fashion MNIST 4CNN non-i.i.d.

| Method | Acc (mean ± std) | bpp | bpp (BC) | Uplink | Downlink |
|---|---|---|---|---|---|
| FedAvg | $0.867 \pm 0.1$ | 64.0 | 35.0 | 32.0 | 32.0 |
| Doublesqueeze | $0.861 \pm 0.2$ | 2.0 | 1.1 | 1.0 | 1.0 |
| Memsgd | $0.863 \pm 0.2$ | 33.0 | 4.2 | 1.0 | 32.0 |
| Liec | $0.853 \pm 0.1$ | 4.5 | 2.5 | 2.3 | 2.3 |
| Cser | $0.781 \pm 0.1$ | 34.0 | 4.3 | 1.0 | 33.0 |
| Neolithic | $0.864 \pm 0.2$ | 4.0 | 2.2 | 2.0 | 2.0 |
| M3 | $0.782 \pm 0.2$ | 15.0 | 2.2 | 8.0 | 6.9 |
| BiCompFL-GR-Adaptive | $0.866 \pm 0.03$ | 0.21 | 0.04 | 0.021 | 0.19 |
| BiCompFL-GR-Adaptive-Avg | $0.853 \pm 0.04$ | 0.11 | **0.021** | **0.011** | 0.1 |
| BiCompFL-GR-Fixed | $0.868 \pm 0.03$ | 0.31 | 0.059 | 0.031 | 0.28 |
| BiCompFL-GR-Reconst-Fixed | $0.86 \pm 0.02$ | 0.34 | 0.062 | 0.031 | 0.31 |
| BiCompFL-PR-Fixed | $\mathbf{0.869 \pm 0.03}$ | 0.34 | 0.34 | 0.031 | 0.31 |
| BiCompFL-PR-Fixed-SplitDL | $0.831 \pm 0.03$ | **0.062** | 0.062 | 0.031 | **0.031** |

For 6CNN trained on CIFAR-10, the negative effects of missing global shared randomness and reconstructing in the case of BICOMPFL-GR are prominent. For non-i.i.d. data distributions, the adaptive average allocation shows improvements over the fixed or the average block allocation. Partitioning the model is not a viable option in this setting, especially under non-i.i.d. data.

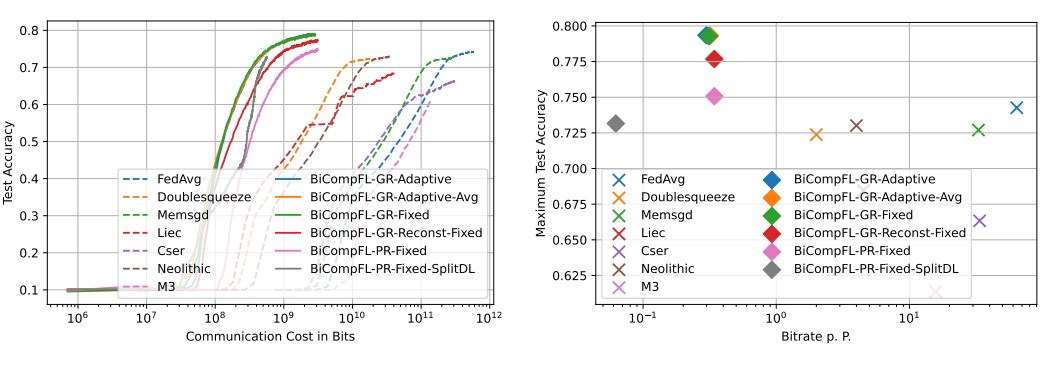

(a) Test Accuracy over Communication

(b) Test Accuracy over Bitrate

Figure 9: CIFAR-10 6CNN i.i.d.

Table 11: CIFAR-10 6CNN i.i.d.

| Method | Acc (mean ± std) | bpp | bpp (BC) | Uplink | Downlink |
|---|---|---|---|---|---|
| FedAvg | $0.742 \pm 0.1$ | 64.0 | 35.0 | 32.0 | 32.0 |
| Doublesqueeze | $0.723 \pm 0.1$ | 2.0 | 1.1 | 1.0 | 1.0 |
| Memsgd | $0.727 \pm 0.1$ | 33.0 | 4.2 | 1.0 | 32.0 |
| Liec | $0.684 \pm 0.09$ | 4.5 | 2.5 | 2.3 | 2.3 |
| Cser | $0.663 \pm 0.08$ | 34.0 | 4.3 | 1.0 | 33.0 |
| Neolithic | $0.73 \pm 0.1$ | 4.0 | 2.2 | 2.0 | 2.0 |
| M3 | $0.614 \pm 0.1$ | 16.0 | 2.2 | 8.3 | 7.5 |
| BiCompFL-GR-Adaptive | $\mathbf{0.793 \pm 0.002}$ | 0.3 | **0.057** | **0.03** | 0.27 |
| BiCompFL-GR-Adaptive-Avg | $0.793 \pm 0.002$ | 0.32 | 0.061 | 0.032 | 0.29 |
| BiCompFL-GR-Fixed | $0.793 \pm 0.004$ | 0.31 | 0.059 | 0.031 | 0.28 |
| BiCompFL-GR-Reconst-Fixed | $0.777 \pm 0.002$ | 0.34 | 0.062 | 0.031 | 0.31 |
| BiCompFL-PR-Fixed | $0.751 \pm 0.003$ | 0.34 | 0.34 | 0.031 | 0.31 |
| BiCompFL-PR-Fixed-SplitDL | $0.732 \pm 0.02$ | **0.062** | 0.062 | 0.031 | **0.031** |

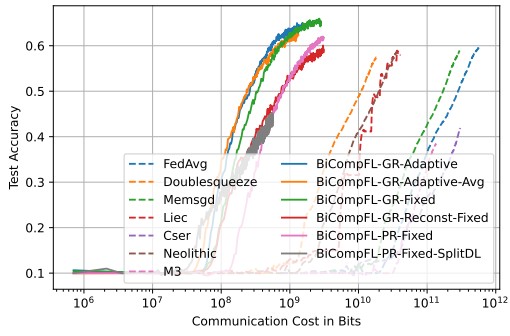

(a) Test Accuracy over Communication

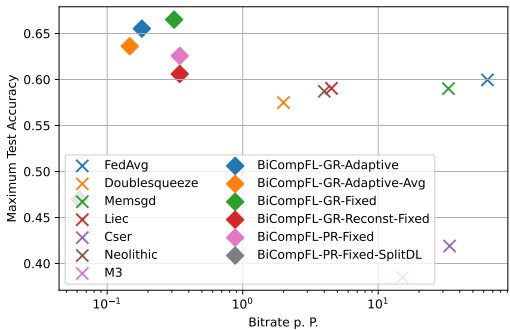

(b) Test Accuracy over Bitrate

Figure 10: CIFAR-10 6CNN non-i.i.d.

Table 12: CIFAR-10 6CNN non-i.i.d.

| Method | Acc (mean ± std) | bpp | bpp (BC) | Uplink | Downlink |
|---|---|---|---|---|---|
| FedAvg | $0.599 \pm 0.1$ | 64.0 | 35.0 | 32.0 | 32.0 |
| Doublesqueeze | $0.575 \pm 0.1$ | 2.0 | 1.1 | 1.0 | 1.0 |
| Memsgd | $0.589 \pm 0.1$ | 33.0 | 4.2 | 1.0 | 32.0 |
| Liec | $0.589 \pm 0.2$ | 4.5 | 2.5 | 2.3 | 2.3 |
| Cser | $0.419 \pm 0.09$ | 34.0 | 4.3 | 1.0 | 33.0 |
| Neolithic | $0.587 \pm 0.1$ | 4.0 | 2.2 | 2.0 | 2.0 |
| M3 | $0.385 \pm 0.1$ | 15.0 | 2.2 | 8.3 | 6.7 |
| BiCompFL-GR-Adaptive | $0.655 \pm 0.04$ | 0.18 | 0.034 | 0.018 | 0.16 |
| BiCompFL-GR-Adaptive-Avg | $0.636 \pm 0.05$ | 0.15 | **0.028** | **0.015** | 0.13 |
| BiCompFL-GR-Fixed | $\mathbf{0.665 \pm 0.03}$ | 0.31 | 0.059 | 0.031 | 0.28 |
| BiCompFL-GR-Reconst-Fixed | $0.606 \pm 0.05$ | 0.34 | 0.062 | 0.031 | 0.31 |
| BiCompFL-PR-Fixed | $0.626 \pm 0.03$ | 0.34 | 0.34 | 0.031 | 0.31 |
| BiCompFL-PR-Fixed-SplitDL | $0.47 \pm 0.07$ | **0.062** | 0.062 | 0.031 | **0.031** |

For completeness, we present in Fig. 11 the test accuracies over the number of trained epochs for all scenarios considered above. The setting of interest to this work is that of limited communication cost, and in particular, which performance is achievable given a fixed communication budget. Nonetheless, we can find that our proposed methods are not inferior in convergence speed over epochs compared to the baselines.

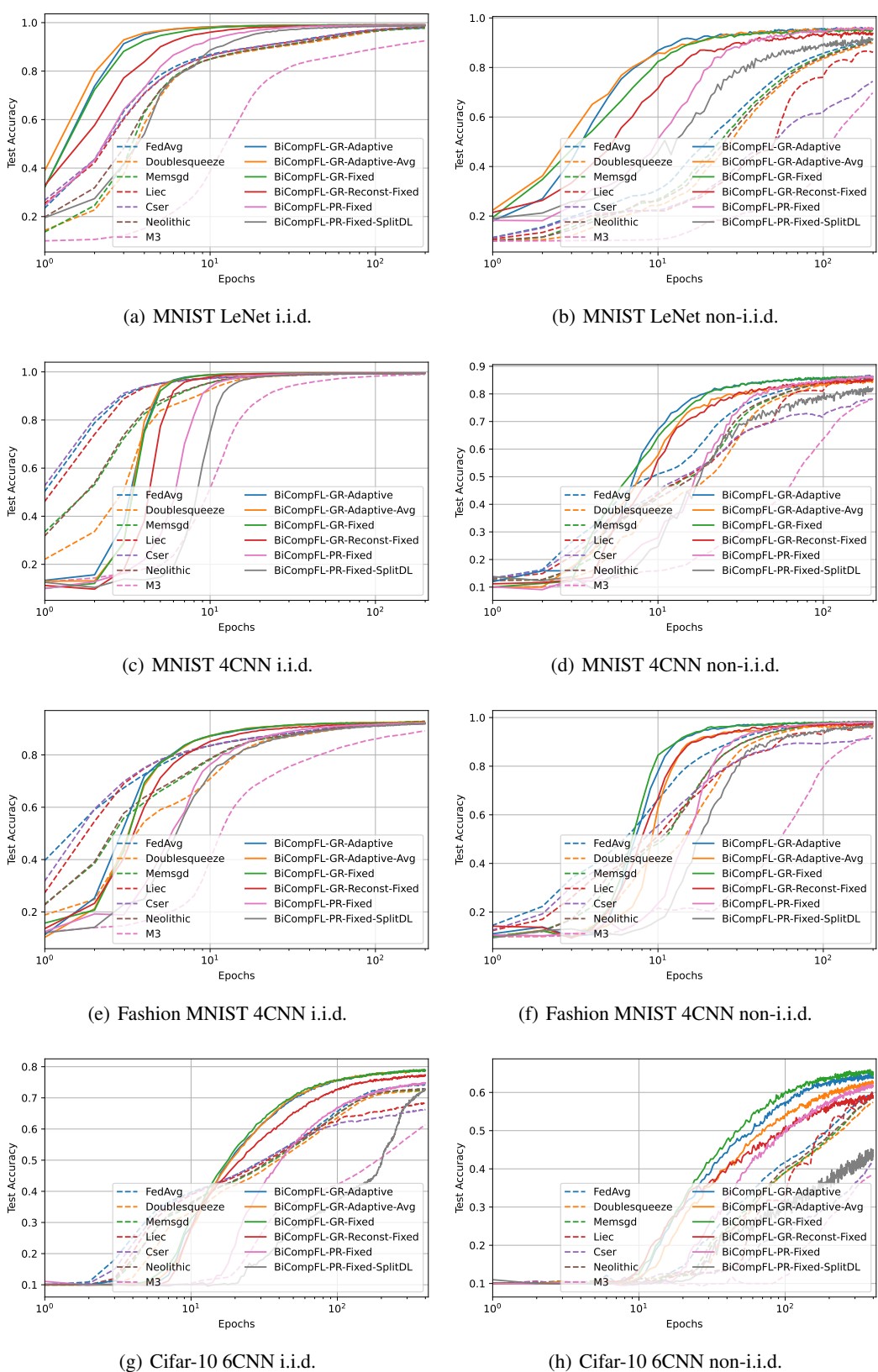

(a) MNIST LeNet i.i.d.

(b) MNIST LeNet non-i.i.d.

(c) MNIST 4CNN i.i.d.

(d) MNIST 4CNN non-i.i.d.

(e) Fashion MNIST 4CNN i.i.d.

(f) Fashion MNIST 4CNN non-i.i.d.

(g) Cifar-10 6CNN i.i.d.

(h) Cifar-10 6CNN non-i.i.d.

Figure 11: Test Accuracy over Epochs

# I Ablation Studies

## I.1 Number of Clients

We study in what follows the sensitivity to various hyperparameters of our algorithms. For comparability, we conduct all experiments on the model 4CNN, Fashion MNIST, and i.i.d.data. We plot for all experiments the accuracies over the number of epochs, and over the communication cost in bits. We first evaluate in Fig. 12 the effectiveness of BICOMPFL-PR and BICOMPFL-PR for different numbers of clients. It can be found that both algorithms exhibit satisfying performance even for $n = 50$, given that the same data is now distributed on more clients. The overall communication cost increases by roughly the factor of the increase in the number of $n$. To illustrate this further, we additionally plot in Fig. 13 the bitrates per parameter.

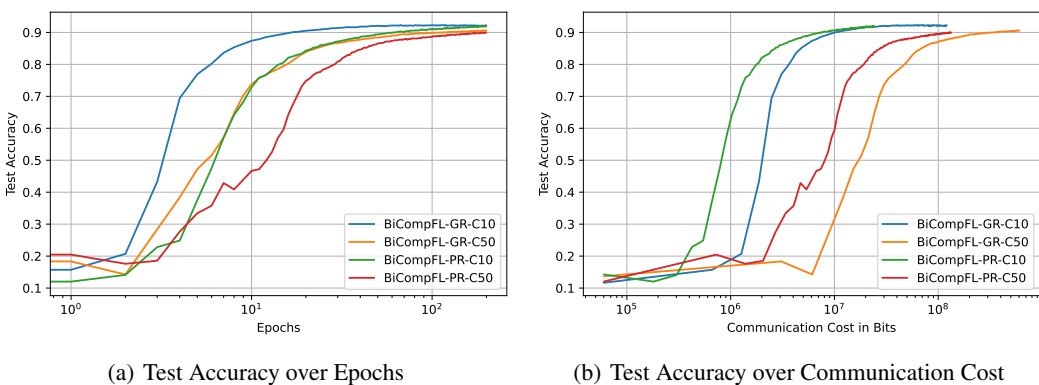

(a) Test Accuracy over Epochs   (b) Test Accuracy over Communication Cost

Figure 12: BICOMPFL-GR and BICOMPFL-GR With Different Number of Clients

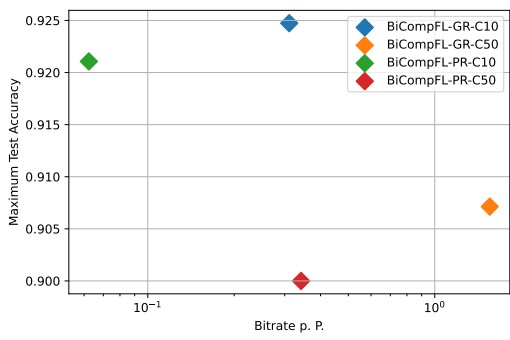

Figure 13: Bitrates for BICOMPFL-GR and BICOMPFL-GR With Different Number of Clients

## I.2 Optimization of the Prior

As described in the main body of the paper, BICOMPFL-PR allows for optimizing the choice of the prior at the clients by optimizing the convexity parameter $\lambda$ that mixes the global model estimate with the posterior transmitted by the client an iteration ahead, i.e., $p_{i,u}^t = \lambda \hat{\theta}_{i,t} + (1 - \lambda)\hat{q}_i^t$ to reduce the communication cost. To evaluate the potential of this method, we optimize $\lambda$ so that it minimized the KL-divergence between the current posterior $q_i^t$ (to be transmitted) and the prior $p_{i,u}^t$, representative for the uplink communication cost. The KL-minimizing $\lambda$ is transmitted to the federator, which is necessary for the federator to reconstruct the importance samples. This optimization is conducted at each iteration individually at the clients. We present in Fig. 14 the performance of this method compared with the algorithms that use as priors exclusively the global model estimates of the clients. Note that optimizing the prior individually at the clients is only possible for BICOMPFL-PRWe plot

the performance of BICOMPFL-GR for reference only. To assess the potential, we ignore for the moment the cost of transmitting $\lambda$, which could be reduced by further compression techniques and leveraging the inter-round dependencies of the choice of $\lambda$.

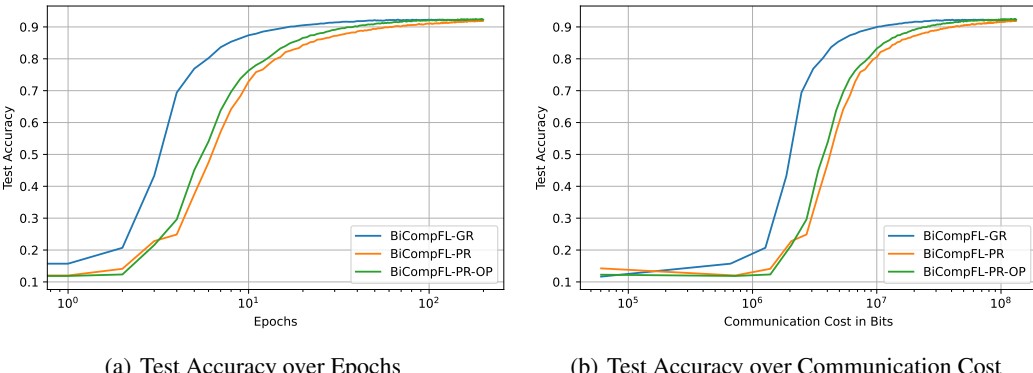

(a) Test Accuracy over Epochs     (b) Test Accuracy over Communication Cost

Figure 14: BICOMPFL-PR With and Without Optimization over the Prior. Optimization over the Priors is denoted by OP.

It can be found that, while optimizing the prior improves the accuracy over epochs and with respect to the communication cost compared to BICOMPFL-PR the improvements are rather insignificant. We therefore present for clarity the algorithm with a fixed choice of the prior as the former global model estimate, which additionally reduce the computation overhead at the clients by avoiding the optimization over $\lambda$. Nonetheless, we note that in certain edge cases, there can be merit in the optimization approach, for instance when the number $n_{\mathrm{DL}}$ of samples on the downlink is very small, and hence the global model estimate is inaccurate.

### I.3   Number of Samples

We continue to assess the impact of the number $n_{\mathrm{DL}}$ of samples on the downlink. We therefore evaluate the performance of BICOMPFL-PR for $n_{\mathrm{DL}} \in \{5, 10, 20\}$. We evaluate the differences on BICOMPFL-PR The results in Fig. 15 reflect the obvious: the larger $n_{\mathrm{DL}}$, the better the accuracy when plotted over the number of epochs. On the contrary, the larger $n_{\mathrm{DL}}$, the larger the communication cost per epoch. The final accuracies do not show substantial differences, and hence, $n_{\mathrm{DL}} = 5$ is sufficient in this setting. To avoid assessing our method overly optimistic and provide a fair comparison to other methods, we choose $n_{\mathrm{DL}} = 10$ in all our experiments, noting that the communication can further be reduced in certain scenarios by lowering $n_{\mathrm{DL}}$ without notable performance loss.

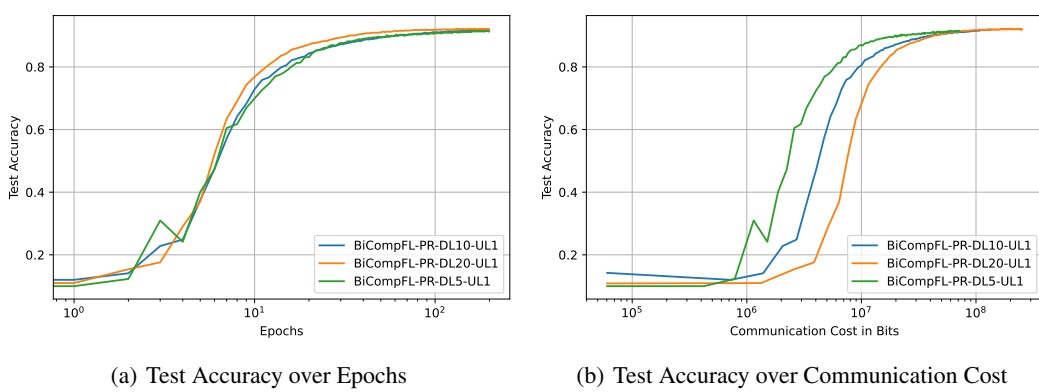

(a) Test Accuracy over Epochs     (b) Test Accuracy over Communication Cost

Figure 15: BICOMPFL-PR for Different Number of Downlink Samples and a Single Uplink Sample.

### I.4 Block Size

We compare in Fig. 16 the performance of BICOMPFL-GR for different block sizes BS $= d/B \in \{128, 256, 512\}$. As expected, fixing $n_{\mathrm{IS}}$, larger block sizes worsen the performance of the algorithm when evaluated over the number of epochs. However, larger block sizes simultaneously reduce the communication cost, and can hence be beneficial in many scenarios. However, we also note that larger block sizes comes at the expense of increases sampling complexities, and hence, the maximum block sizes are also dominated by the resources of the clients and the federator.

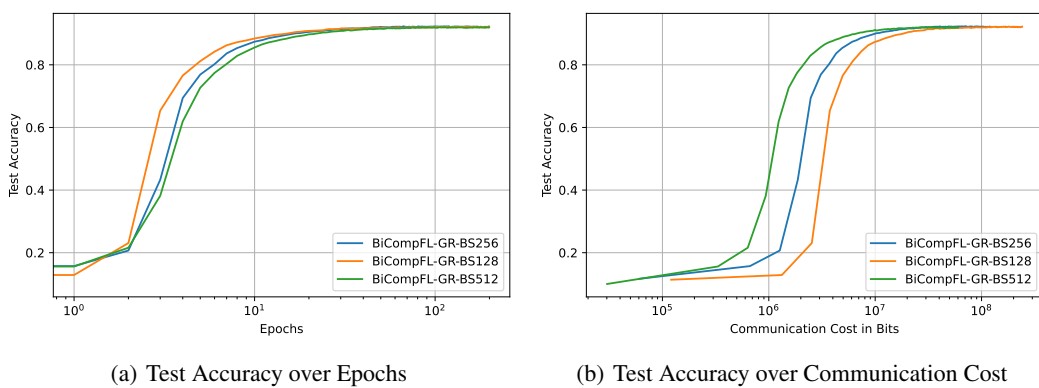

(a) Test Accuracy over Epochs        (b) Test Accuracy over Communication Cost

Figure 16: BICOMPFL-GR With Fixed Block Allocation for Varying Block Sizes (BS) $d/B$.

### I.5 Number of Importance Samples

In Fig. 17, we study the sensitivity of our algorithms with respect to the number of importance samples $n_{\mathrm{IS}}$ at the example of BICOMPFL-GR. While larger number of $n_{\mathrm{IS}}$ slightly improves the performance as of the epoch number, the improvements do not outweigh the additional communication costs. Overall, our algorithm proves rather stable within reasonable ranges for $n_{\mathrm{IS}}$. We fix in all our experiments $n_{\mathrm{IS}} = 256$, presenting a good trade-off between performance and efficiency.

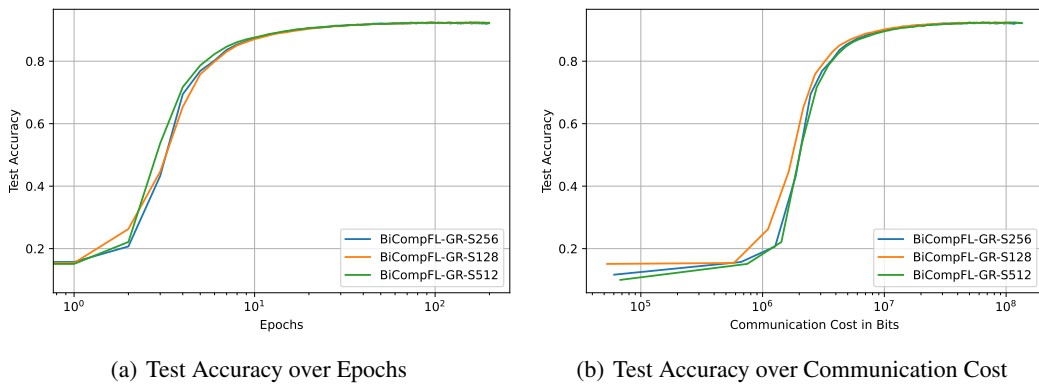

(a) Test Accuracy over Epochs        (b) Test Accuracy over Communication Cost

Figure 17: BICOMPFL-GR with Varying Number of Importance Samples $n_{\mathrm{IS}}$ per Block.

### I.6 Learning Rate

Our main claims are centered around the per-client bitrates per parameter, rendering the choices of learning rate secondary to our reasoning. Nonetheless, we tune the learning rates of all methods so that the baselines and BICOMPFL achieve roughly the same final accuracies, allowing a fair comparison of resulting communication costs. We analyze the impact of the learning rate choice

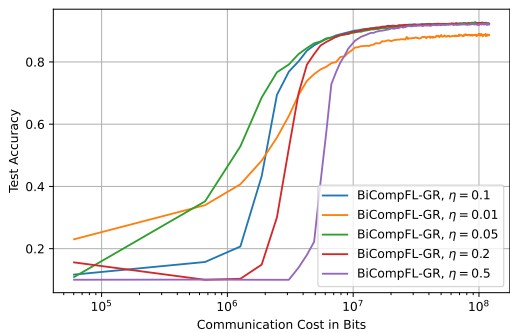

Figure 18: BiCompFL-GR with Varying Number of Importance Samples $n_{IS}$ per Block.

on BiCompFL in Fig. 18, for $\eta \in \{0.01, 0.05, 0.1, 0.2, 0.5\}$. It is particularly noteworthy that BiCompFL exhibits stable performance across most learning rates we study, which we attribute to the regularization effects that occur in stochastic FL, detailed in the main body of the paper. Only for $\eta = 0.01$, the final performance is decreased, indicating that BiCompFL is not able to escape local optima in this setting. Although $\eta = 0.05$ provides the best communication efficiency, we choose a moderate learning rate of $\eta = 0.1$ not to overestimate our method compared to other approaches.

