# OpenReview forum: "Bi-Directional Communication-Efficient Stochastic FL via Remote Source Generation"
_NeurIPS.cc/2025/Conference — NeurIPS 2025 poster_

### Official Review · Reviewer_Bbin · 2025-07-02

**Clarity:** 4
**Significance:** 3
**Originality:** 3
**Rating:** 5
**Confidence:** 4

**Summary:**

The paper considers stochastic federated learning to reduce communication costs in both uplink and downlink communication. The techniques are based on Minicam random coding, an existing remote source generation tool. Both theoretical guarantees and empirical validations are provided.

**Questions:**

I have a question about the choice of the remote source generation scheme. The authors used Minicam random coding, which is non-exact: the output distribution is distorted in the sense that the remotely generated sample only follows $\hat{Q}$, rather than $Q$ exactly.I believe the authors are aware of this (as noted in lines 96–97). My question is whether the authors have considered exact schemes, such as the Poisson functional representation by Li and El Gamal (2018) or dithered-quantization-based schemes. It is fine if these were not considered, I'm just curious whether exactness might make a difference in FL tasks, for example in terms of accuracy.

**Ethical Concerns:**

["NO or VERY MINOR ethics concerns only"]

**Final Justification:**

I will keep my score.

**Limitations:**

There are no new limitations from my side. The authors have already discussed some limitations, e.g.,  the availability of shared randomness and some required assumptions.

**Paper Formatting Concerns:**

This is not major formatting issues from my perspective.

**Quality:**

4

**Strengths And Weaknesses:**

** Strengths**:

- The paper is well written and well structured.
- The contribution is clear, and the literature review is thorough.
- The work makes a solid contribution, both from theoretical and experimental perspectives.

** Weaknesses**:

There are no obvious major weaknesses. I would suggest the author consider also trying other remote source generation schemes to see if any of them achieve better results in FL tasks.

---

> ### Author Rebuttal · Authors · 2025-07-31
>
> We thank Reviewer Bbin for their positive evaluation of our work and for their thoughtful comments. We answer their question as follows.
>
> - **Q1/W1** (Other remote generation schemes) The choice of MRC is motivated mainly by practicality and ease of exposition. A natural improvement is ordered random coding (ORC) [R1], which reduces the entropy of the indices selected for transmission using the Gumbel-Max trick. This extension can directly be applied in our method with a minor adaption that we refrained from including here for clarity. On a related note, the authors of ORC [R1] further expose an interesting connection to the Possion functional representation mentioned by the reviewer. While such proposals allow exact remote generation, lossless sampling schemes usually incur substantially larger communication costs and sampling complexities, often rendering those methods impractical. We believe that the overhead required for lossless compression compared to the lossy variants used in our methods are insufficiently motivated in a federated learning context, since the variance of the training process likely outweighs the negative effects of the noise of the sampling process conducted for compression. If the reviewer deems it useful, we are happy to add a respective discussion in the paper.
>
> [R1] L. Theis and N. Yosri, “Algorithms for the communication of samples,” in International Conference on Machine Learning, 2022, pp. 21 308–21 328.

---

> ### Comment · Reviewer_Bbin · 2025-08-02
>
> Dear all,
>
> Thank you for your clarification, and I appreciate your work. I do not have further questions.
>
> I have also read the comments (especially the weaknesses) from other reviewers. I agree that larger-scale experiments are preferable, and that the availability and practicality of shared randomness might be a weakness (which the authors have addressed in the paper and responded to in the rebuttal). Adding some discussion on privacy and fairness would also be beneficial. From my personal perspective, introducing remote source generation to FL and demonstrating academic-level empirical validation are already meaningful contributions. I also find the writing quality of this paper to be good. I intend to keep my positive rating.
>
> Best regards,
>
> Reviewer Bbin

---

### Official Review · Reviewer_GJUK · 2025-07-03

**Clarity:** 3
**Significance:** 4
**Originality:** 3
**Rating:** 5
**Confidence:** 3

**Summary:**

The authors proposed a bidirectional compression method for a wild federated learning setting in which uplink and downlink communications are both constrained. The underlying motivation is connected to the Bayesian federated learning mechanism, which employs an alternative view that the masks of weights are learned.

**Questions:**

Please refer to the *weaknesses* subsection above.

**Ethical Concerns:**

["NO or VERY MINOR ethics concerns only"]

**Final Justification:**

The authors clearly addressed concerns I raised and no further clarification is required.

**Limitations:**

yes

**Paper Formatting Concerns:**

Please consider unifying citation style to e.g., [XXX et al., 2000] for better readability.

**Quality:**

4

**Strengths And Weaknesses:**

### Strengths
---
*  The paper is well-written and the motivation and algorithm design are both convincing. There are tons of additional analyses that thankfully address most of my preliminary concerns raised during reading the manuscript.
* The consideration of uplink bottlenecks and global/private randomness, both of which are overlooked in academic papers, is impressive.
* The distributional convergence of a global model is sound and the analysis of communication cost related to the divergence is also intriguing.
* The empirical validation is conducted extensively on both IID and non-IID scenarios using widely-used benchmark datasets with moderate number of clients.

### Weaknesses
---
* Based on Algorithm 1, the proposed method seems not support partial client pariticpation, which is often required in low-bandwith settings.
* To better illustrate the practical strength and justification of the small number of clients in the experiments, the authors could narrow down the target federation setting to a cross-silo federated learning setting (see Table 1 of Kairouz et al., 2018), or specify/exemplify plausible scenarios in which the proposed methods are most applicable.
* (minor question) Is there any (theoretical) tips to set the number of quantization intervals, $s$?

- (Kairouz et al., 2018) Advances and Open Problems in Federated Learning

---

> ### Author Rebuttal · Authors · 2025-07-31
>
> We thank Reviewer GJUK for their positive evaluation of our work. We address their concerns as follows.
>
> - **W1** (Partial client participation) Although we do not consider partial client participation in the paper, our methods are compatible with such extensions. For BiCompFL-PR, partial client participation can readily be implemented on top of the methods we propose, as they do not rely on full prior synchronization among the clients. BiCompFL-GR on the other hand requires that all clients' priors are synchronized at each iteration. To allow partial client participation, the federator is required to transmit the previous global model to the clients that were absent in the previous iteration. This ensures full synchronization of the global model estimate used for communication-efficient uplink communication. Hence, while partial client participation is supported by BiCompFL-PR without additional communication overhead, BiCompFL-GR's compatibility with partial client participation comes with a potentially increased downlink communication cost for clients that were previously passive. If the reviewer deems it helpful, we will be happy to add a discussion on this in the paper.
> - **W2** (Practical strengths and justification of small number of clients): The number of clients is not an inherent limitation in our methods. While we experimentally examined our methods with at most 50 clients, our methods exhibit favorable scaling properties in the number of clients. The more clients, the better the averaging effects of the noise introduced by the sampling operations for uplink compression, effectively reducing the error made in estimating the clients' average posterior. We selected the number of clients used in the experiments in alignment with common studies in the literature and our available resources.
> - **W3** (Practical tips for the choice of quantization intervals): In general, the choice of the number of quantization intervals depends on the model architecture and dataset complexities, affecting the number of iterations required until convergence. The simpler the learning task, the fewer quantization intervals are sufficient for convergence. In addition to generic arguments, we particularly want to highlight that the number of quantization levels should be carefully selected respecting the expected variance of the gradients. If gradients exhibit inherently large variance, e.g., through small mini-batch sizes, large variance of the local datasets, or substantially over-parameterized networks, small number of quantization intervals might be particularly efficient, and increasing that number would only negligibly improve the performance. We also want to remark that one-bit quantization was often shown to be trivially effective [R1, R2]. We hope these explanations provide a glimpse on the problem of quantization interval selection.
> - **Formatting**: We thank the reviewer for this comment and will unify the citation style accordingly.
>
> [R1] F. Seide, H. Fu, J. Droppo, G. Li, and D. Yu, “1-bit stochastic gradient descent and its application to data-parallel distributed training of speech DNNs,” in Interspeech, 2014, pp. 1058–1062.
>
> [R2] S. P. Karimireddy, Q. Rebjock, S. Stich, and M. Jaggi, “Error feedback fixes SignSGD and other gradient compression schemes,” in International Conference on Machine Learning, vol. 97, 2019, pp. 3252–3261.

---

> > ### Comment · Reviewer_GJUK · 2025-08-01
> >
> > Dear authors, thanks for your clarification.
> > To put a bit more on the cross-silo setup, it can easily justify the current full synchronization requirement, as well as a small number of clients, without further modification. This is because we can naturally assume a moderate number of reliable and addressable clients, under cross-silo FL setting (cf., Kairouz et al., 2018). Thus, I hoped that the cross-silo framing would help clarify the current writing. Though I don't wish to impose, I'd appreciate it if you would ruminate on my original intention.
> >
> > For the quantization interval, it should be carefully tuned according to the task-specific constraints, right?
> > If so, please consider adding a line noting which factors are critical to the tuning process and should be prioritized.
> > Thank you.
> >
> > Best,
> > Reviewer GJUK

---

> > > ### Author Response · Authors · 2025-08-06
> > >
> > > We thank the Reviewer for emphasizing this helpful thought, which further helps in positioning our paper. We are happy to add such a phrasing in the manuscript as to why our framework, among others, is particularly applicable in the cross-silo FL setting.
> > >
> > > Regarding quantization, the Reviewer rightfully summarizes our explanations. The number of quantization intervals should indeed be tuned based on task-specific constraints. As suggested by the reviewer, we will add a respective explanation to the paper.

---

> > > > ### Comment · Reviewer_GJUK · 2025-08-06
> > > >
> > > > Thank you for considering my suggestions. Fingers crossed!

---

### Official Review · Reviewer_oHgr · 2025-07-03

**Clarity:** 3
**Significance:** 3
**Originality:** 3
**Rating:** 5
**Confidence:** 4

**Summary:**

This paper addresses the significant communication overhead in both uplink (client-to-server) and downlink (server-to-client) directions for ​**stochastic (Bayesian) Federated Learning (FL)​**, where probabilistic models are trained to capture parameter uncertainty. Unlike prior work focused on deterministic FL or unidirectional compression, the authors propose ​**BICOMPFL**, a novel framework leveraging ​**remote source generation**​ via ​**Minimal Random Coding (MRC)​**​ to enable bidirectional communication efficiency.

The core insight exploits shared randomness and ​**side information**​ (in the form of prior distributions) to allow clients and the server to sample from each other's local/global posteriors rather than transmitting explicit updates. This transforms communication into a sampling problem, where transmission costs scale with the KL-divergence between updated distributions and their priors. Two algorithm variants are introduced: ​**BICOMPFL-GR**​ (using global shared randomness) enables efficient relay-based compression by synchronizing priors across all parties, while ​**BICOMPFL-PR**​ (using private randomness per client-server pair) handles settings without globally synchronized randomness, albeit with slightly higher overhead.

Key contributions include:
1) ​**Inherent communication regularization**​ through a mirror-descent optimization structure, treating communication costs as part of the learning objective.
2) ​**Substantial empirical gains**: Experiments on MNIST, Fashion MNIST, and CIFAR-10 demonstrate ​**5–32× reduction in total communication costs**​ (bits per parameter) while maintaining competitive accuracy against strong baselines (e.g., FedAvg, Doublesqueeze, M3). Adaptive block allocation strategies further optimize costs.
[](@replace=1)

3) ​**Theoretical innovations**: Refined bounds for MRC compression (for Bernoulli distributions) quantify uplink/downlink trade-offs, and a new contraction property is proven for biased MRC operators, enabling convergence guarantees in conventional FL settings.
4) ​**Methodological generality**: Extension to conventional FL via stochastic quantization (e.g., SignSGD) shows broader applicability without error feedback mechanisms.

The work bridges stochastic FL with practical communication constraints, demonstrating that principled probabilistic modeling and remote sampling can dramatically reduce bidirectional bandwidth while preserving model performance.

**Questions:**

​**1. Scalability Validation to Modern Architectures**​
The experimental validation relies exclusively on small CNNs (LeNet, 4-6 layer CNNs). To strengthen claims of broader applicability, please evaluate BICOMPFL on larger models (e.g., ResNet-18/50, ViT-Tiny) and complex datasets (e.g., ImageNet subset or CIFAR-100). Specifically, quantify how communication costs scale with model size and heterogeneity under your framework.

​**2. Generalization of Theoretical Guarantees**​
The theoretical analysis (Section 5) focuses entirely on Bernoulli distributions for posterior/prior modeling. Clarify whether Lemma 1’s contraction property and Theorem 1’s communication bounds extend to other distributions common in Bayesian FL (e.g., Gaussian posteriors). Provide either a proof sketch for multivariate Gaussians or empirical validation via Gaussian-based stochastic FL experiments. Should these extensions hold, emphasize their implications in the text; if not, explicitly state the limitations of the current theory and propose mitigation strategies (e.g., parameter transformation techniques).

​**3. Algorithm Workflow Clarity**​
The distinction between BICOMPFL-GR and BICOMPFL-PR is not intuitively conveyed in the pseudocode (Algorithms 1-2). Consider adding a schematic diagram illustrating the bidirectional remote sampling workflow, highlighting critical differences in prior synchronization, index relaying, and noise propagation. [](@replace=1)
 demonstrates performance gaps but lacks visual workflow context. A figure showing how shared randomness reduces downlink variance would clarify why BICOMPFL-GR outperforms BICOMPFL-PR.

​**4. Privacy and Fairness Implications**​
The paper does not address how compressed bidirectional updates affect privacy vulnerabilities (e.g., membership inference/gradient inversion attacks) or fairness across heterogeneous clients. Analyze:
- How MRC-based sampling compares to DP-SGD in privacy-utility tradeoffs
- Whether adaptive block allocation biases aggregation against clients with high KL-divergence
- Test accuracy variance across demographic groups in non-i.i.d. settings
Including a "Broader Impacts" section addressing these points would align with NeurIPS ethics guidelines and strengthen societal relevance.

**Ethical Concerns:**

["NO or VERY MINOR ethics concerns only"]

**Final Justification:**

Most of our concerns are resolved during the rebuttal and discussion phase. The paper's core innovations are now rigorously supported and accessible. With these enhancements implemented in the camera-ready version, we will increase our score.

**Limitations:**

See weaknesses.

**Paper Formatting Concerns:**

No major formatting issues.

**Quality:**

4

**Strengths And Weaknesses:**

### Strengths
The paper demonstrates ​**exceptional technical quality**​ through rigorous theoretical development and comprehensive experimental validation. The introduction of ​**BICOMPFL**​ represents a significant advancement in stochastic federated learning, with novel bidirectional compression mechanisms based on Minimal Random Coding (MRC) and remote source generation. The theoretical contributions are particularly strong, including refined MRC bounds for Bernoulli distributions and a new contraction property for biased compressors (Lemma 1), which enables convergence guarantees in conventional FL settings. Experimentally, the work is thorough, validating claims across three datasets (MNIST, Fashion MNIST, CIFAR-10) and both i.i.d. and non-i.i.d. settings, consistently demonstrating ​**5–32× communication reduction**​ without accuracy degradation. The adaptive block allocation strategies and extension to conventional FL showcase methodological versatility. The significance is high, addressing a critical bottleneck in FL deployment—communication overhead—through a principled probabilistic framework that inherently treats communication as an optimization objective via mirror descent. The approach is highly original, being the first to solve bidirectional compression in stochastic FL through remote sampling and prior-based side information. [](@replace=1)
 visually encapsulates this breakthrough by showing order-of-magnitude bitrate improvements over baselines.

### Weaknesses
While technically solid, the paper has ​**clarity issues**​ in theoretical sections where derivations become overly dense (e.g., Lemma 2 proof in Appendix A). The distinction between BICOMPFL-GR and BICOMPFL-PR could be better motivated visually—a conceptual diagram would help readers grasp the remote generation workflow. Experimentally, the ​**quality is slightly compromised by limited scope**: Evaluations use only small CNNs rather than modern architectures (e.g., ResNets), and comparisons exclude recent Bayesian FL methods like FedBE. The significance claims about scalability lack validation beyond 50 clients, and energy/runtime metrics—critical for edge deployment—are absent. The theoretical ​**originality is constrained to Bernoulli distributions**, with no discussion of extending MRC analysis to Gaussian posteriors common in Bayesian FL. The ​**significance of broader impacts**​ is underdeveloped: While reduced communication benefits sustainability, potential privacy risks from compressed updates (e.g., membership inference) aren't addressed. [](@replace=2)
 reveals another weakness: convergence instability in BICOMPFL-PR under high heterogeneity, which isn't sufficiently analyzed. Finally, adaptive block allocation's overhead isn't quantified, leaving practical trade-offs unclear.

---

> ### Author Rebuttal · Authors · 2025-07-31
>
> We thank Reviewer oHgr for their comments. We address the reviewer's concerns as follows.
>
> - **W1** (Clarity issues): We are unsure which parts are referred to as overly dense. If the reviewer has particular pointers, we are happy to modify the passages accordingly.
>
> - **W2/Q3** (Algorithm workflow clarity): We are happy to add such a conceptual diagram to the paper highlighting the differences between the two proposed methods, which lie mostly in the way shared randomness is leveraged.
>
> - **W3/Q1** (Scalability validation to model architectures):
> Our work proposes a novel method within the scope of academic research. Given standard resources, we conducted experiments with complexities that are comparable to all related works in this line of research.Given the numerical and theoretical evidence, we could not identify any particular negative effects as the task complexities scale; on the contrary, the positive effects of substantially reduced communication complexity gain increasing importance.
> The communication costs are quantified both in theory and experiments for varying model sizes and levels of heterogeneity. We kindly refer the reviewer to the respective sections and various experiments, also in the appendix.
>
> - **W4** (Runtime and energy costs): Like other papers in this line of work, we propose a conceptual framework that uses standard models and building blocks, such as sampling from parametric distributions. Many works are particularly concerned with resource-efficient implementations of such generic building blocks, e.g., the sampling operations. To complement our experiments, we theoretically analyze the fundamental sampling complexity of our compression methods. However, implementing and deploying any method on hardware in practice requires additional engineering efforts to make it efficient. Thus, measuring the energy or runtime of our methods on our servers would not yield meaningful comparisons, unless tailoring each method individually to the specific hardware at hand. We, hence, refrain from showing such plots, noting that the results would be orthogonal to the message of our paper.
> Our experiments, which include 50 clientsm align well with the settings considered in the literature. Importantly, we substantially benefit from increased number of clients on the uplink through positive averaging effects of the noise introduced in the sampling operations. Hence, the averaged posterior estimates observed by the federator improve as the number of clients increases. We thus believe that our ablation studies provide sufficient evidence of the positive scaling properties of our method.
>
> - **W5/Q2** (Generalization of theoretical guarantees): We deliberately choose to conduct a tailored analysis for Bernoulli distributions. While it is possible to run a more general analysis for other classes of distributions (as detailed in the sequel), we found that the specific optimization method proposed in the paper is particularly communication-efficient and allows for inherent communication-regularized training, a unique property in this line of work that renders our methods substantially more communication-efficient while preserving state-of-the-art performance.
>
> Nonetheless, our analysis can indeed be extended to other parametric distributions, such as multi-variate Gaussians. The key ingredient is to replace the bias bound in Lemma 2 with the generic results for MRC known from [R1]. Although weaker than our result, their result holds for all classes of distributions. Having such an upper bound on the bias, one can follow our proof strategy to prove the communication costs in Theorem 1, i.e., using the convexity of KL divergence and decomposing the error in parameter estimation into a bias term (where the result from [R1] generically applies) and employing concentration bounds, such as the Hoeffding inequality for subGaussian random variables, to bound the sampling error. The remaining steps follow analogously to our proof. As for the contraction property in Lemma 1 used to establish convergence guarantees for the case of conventional FL, similar adaptations can be made. However, we note that for the composition of stochastic quantization with MRC to turn conventional FL into a stochastic setting, the incentive to use distributions other than Bernoulli might be weaker than for the natively stochastic settings mentioned by the reviewer. We are happy to include a discussion related to this in the paper.
>
> - **W6/Q4** (Privacy and fairness implications): Although privacy and fairness are issues  widely studied in the literature, this work does not primarily focus on those aspects. For instance, privacy in FL with relative entropy coding was considered in [R2]. On the other hand, as a general statement, compression methods positively affect the clients' privacy in FL, which mainly stems from the fundamental data processing inequality [R3]. In our case, the noise introduced throughout the sampling operations used for compression dilutes the individual contributions of the clients observed by the federator, thus naturally enhancing the clients' privacy and reducing the risk of privacy breaches through, e.g., membership inference attacks. Concerning fairness, we note that adaptive block allocation does not have implications on potential bias w.r.t. particular clients in the aggregation process. The same amount of samples are observed for each client, ensuring balance in the aggregation process. Further testing the particularities of variance with regard to demographic groups is beyond the scope of this work. However, we are happy to add a respective discussion in our paper.
>
> - **W7** (Convergence instability under high heterogeneity): We kindly point the reviewer to our extensive experiments in the appendix, where we examine the stability of our methods with high data heterogeneity modeled by a Dirichlet with parameter $\alpha=0.1$, which is considered an extremely heterogeneous regime [R4].
>
> - **W8** (Overhead of adaptive block allocation): All our experiments consider the quantitative cost of adaptive block allocation incurred to transmit the block locations. We find that the overhead for transmitting block indices is negligible compared to the positive effects of block size adaptation on the communication cost. As the training progresses, adaptive partitioning successively increases the block sizes, thus drastically reducing the communication costs over the runtime of the algorithm.
>
> [R1] S. Chatterjee and P. Diaconis, “The sample size required in importance sampling,” The
> Annals of Applied Probability, vol. 28, no. 2, pp. 1099–1135, 2018.
>
> [R2] A. Triastcyn, M. Reisser, and C. Louizos, DP-REC: Private and communication-efficient federated learning, 2022.
>
> [R3] T. Cover and J. A. Thomas, Elements of information theory. Wiley-Interscience, 2006.
>
> [R4] Y. Allouah, S. Farhadkhani, R. Guerraoui, N. Gupta, R. Pinot, and J. Stephan, “Fixing by
> mixing: A recipe for optimal Byzantine ML under heterogeneity,” in International Conference
> on Artificial Intelligence and Statistics, 2023, pp. 1232–1300.

---

### Official Review · Reviewer_EDwN · 2025-07-21

**Clarity:** 3
**Significance:** 2
**Originality:** 3
**Rating:** 3
**Confidence:** 4

**Summary:**

The paper proposes BICOMPFL, a family of federated learning (FL) algorithms that cut both uplink and downlink communication by exploiting the stochastic/Bayesian view of model parameters and using Minimal Random Coding (MRC) with shared priors (“side information”) to remotely generate samples at the receiver instead of transmitting full local samples or model updates. The paper also provides experimental results to show the approach cuts total communication by about 5–32× while maintaining competitive accuracy.

**Questions:**

Questions:
1. Can you provide evaluations results on large-scale benchmarks and models?
2. On-device, how does BICOMPFL’s runtime and energy compare to vanilla FedAvg?
3. When client data are extremely heterogeneous, does the KL gap explode for some clients?
4. Could BICOMPFL be stacked with other sparsification or structured pruning-based compression methods?

**Ethical Concerns:**

["NO or VERY MINOR ethics concerns only"]

**Quality:**

3

**Strengths And Weaknesses:**

Strengths:
1. The paper aims to reduce the communication overhead in FL training and proposes a probabilistic compression method, which is novel to the my knowledge.
2. The paper also provides the convergence rate to show the proposed algorithm leads to similar convergence rate as FedAvg.
3. The paper includes some empirical evaluations and ablation studies to show the effectiveness of the algorithm.

Weakness:
1. Empirical validation limited to small vision benchmarks.
2. Minimal Random Coding needs n_{IS} samples which scales exponentially w.r.t. the KL divergence between the prior and posterior, which makes the high-dimensional models hard to reach the condition.
3. The most communication-efficient variant (BICOMPFL-GR) assumes all clients share global pseudo-randomness, but this shared randomness is hard to maintain in real world FL systems, establishing and maintaining perfectly synchronized randomness can be difficult. (But this is a minor point compared to 1 and 2)

---

> ### Author Rebuttal · Authors · 2025-07-31
>
> We thank Reviewer EDwN for their thorough review and helpful suggestions. We address their concerns as follows.
>
> - **W1/Q1** (Empirical validation limited to small vision benchmarks): Our work proposes a novel method within the scope of academic research. Given standard resources, we conducted experiments with complexities that are comparable to all related works in this line of research. Moreover, through our extensive numerical and theoretical results, we did not identify any particular negative effects as the task complexities scale; on the contrary, the positive effects of substantially reduced communication complexities gain increasing importance with larger models. % Nonetheless, we are currently running larger-scale experiments for follow-up works.
>
> - **W2** (MRC sampling complexity scales exponentially in the KL-divergence): As the theory suggests, the sampling complexity of MRC scales exponentially in the KL-divergence between prior and posterior. To overcome this limitation, we employ the method of block allocation. Therefore, we fix a per-block target KL-divergence and partition the model parameters into blocks to match the desired target divergence. We further highlight that the KL-divergence is not static and decreases as the training progresses, often leading to negligible divergences when advancing in the training. As a result, higher-dimensional models actually support meeting the KL-conditions, as larger dimensionality increases the degrees of freedom for block partitioning. With regard to the scaling properties, partitioning the models into blocks of equal KL-divergences leads to a linear dependence of the sampling complexity on the model size, alleviating concerns regarding exploding sampling costs.
>
> - **W3** (Shared randomness difficult to maintain in real-world FL): We clearly articulate the assumption that common randomness is required, and specifically propose a variant for cases where global randomness among all clients might be difficult to maintain. Nonetheless, the fact that our methods are fully synchronous facilitate easy piggy-backing of reasonably large seeds onto global model updates sent to the clients. This relative cost of this overhead diminishes, and becomes negligible, as the model size increases. In our experiments, the one-time transmission of few-bit seeds is sufficient to achieve state-of-the-art performance with substantially reduced communication costs.
>
> - **Q2** (Runtime and energy costs): Similar to other papers in this line of research, we propose a conceptual framework that uses standard models and building blocks, such as sampling from parametric distributions. Many works are particularly concerned with resource-efficient implementations of such generic building blocks, e.g., the sampling operations. To complement our experiments, we theoretically analyze the fundamental sampling complexity of our compression methods. However, naturally, implementing and deploying any method  on hardware in practice requires additional engineering efforts to make it efficient. Thus, measuring the energy or runtime of our methods on our servers would not yield meaningful comparisons, unless tailoring each method individually to the specific hardware at hand. We, hence, refrain from showing such plots, noting that such statements would be orthogonal to the message of our paper.
>
> - **Q3** (Exploding KL gap for extreme heterogeneity): This is a very good point brought up by the reviewer, which we carefully investigated (although not discussed much in the paper). Firstly, we do conduct our experiments in regimes considered extremely heterogeneous in the literature (Dirichlet distribution with $\alpha=0.1$), e.g., [R1], and did not observe exploding KL gaps in such settings. We initially assumed such an effect would be present and studied whether it can actually be leveraged for improved system design. Assuming the clients' posteriors are moving substantially away from the global model when data is very heterogeneous, communication costs with private randomness would benefit substantially from using the previous posterior of the clients as prior instead of the previous global model. However, we couldn't observe notable positive effects from such individual prior selection. The key reason is the implicit regularization of our optimization methods. Specifically, the mirror descent approach studied in our paper naturally regularizes the posterior optimization w.r.t. the KL-divergence between prior and posterior, avoiding KL blow-ups in the optimization. This positively affects both the variance in local optimization in the clients and the incurred communication costs.
>
> - **Q4** (Compatibility with sparsification or structured pruning-based compression): Yes, our methods are entirely compatible with other compression methods that sparsify or prune the models/gradients before transmission. Applied prior to our compression methods, sparsified models can be readily fed into our compression methods without further adaptations. The pruned parameters/distribution are simply not included in the block selection of model parameters, thereby further reducing the incurred communication costs. We are happy to add a respective remark in the paper.
>
> [R1] Y. Allouah, S. Farhadkhani, R. Guerraoui, N. Gupta, R. Pinot, and J. Stephan, “Fixing by
> mixing: A recipe for optimal Byzantine ml under heterogeneity,” in International Conference
> on Artificial Intelligence and Statistics, 2023, pp. 1232–1300.

---

### Decision · Program_Chairs · 2025-09-17

**Decision:**

Accept (poster)

**Comment:**

The paper investigates the stochastic federated learning problem, with the aim of reducing the communication costs in both uplink and downlink. The main technical approach is minimal random coding, an existing remote source generation tool. Both theoretical guarantees and empirical validations are provided. Overall, this paper is well-written. The main concerns raised by the reviewers are: (i) the numerical experiments are limited to small-scale vision tasks and the extensions to large-scale models seem non-trivial; (ii) the proposed compressed updates may incur privacy risks in practice. Nevertheless, the reviewers agree that this paper has its own merits and the contributions are sufficient. Therefore, I recommend Accept.

The authors are encouraged to incorporate the reviewers’ comments and polish the presentation during preparing the final version.